# LoCoT2V-Bench: Benchmarking Long-Form and Complex Text-to-Video Generation

Xiangqing Zheng [1]   Chengyue Wu [2]   Kehai Chen [1]   Min Zhang [1]

## Abstract

Recent advances in text-to-video generation have achieved impressive performance on short clips, yet evaluating long-form generation under complex textual inputs remains a significant challenge. In response to this challenge, we present LoCoT2V-Bench, a benchmark for long video generation (LVG) featuring multi-scene prompts with hierarchical metadata (e.g., character settings and camera behaviors), constructed from collected real-world videos. We further propose LoCoT2V-Eval, a multi-dimensional framework covering perceptual quality, text-video alignment, temporal quality, dynamic quality, and Human Expectation Realization Degree (HERD), with an emphasis on aspects such as fine-grained text-video alignment and temporal character consistency. Experiments on 17 representative LVG models reveal pronounced capability disparities across evaluation dimensions, with strong perceptual quality and background consistency but markedly weaker fine-grained text-video alignment and character consistency. These findings suggest that improving prompt faithfulness and identity preservation remains a key challenge for long-form video generation. Our code and data are released at https://github.com/XqZeppelinhead0702/LoCoT2V-Bench.

## 1. Introduction

In recent years, the rapid advancement of AI-Generated Content (AIGC) and the popularity of short-form video platforms have accelerated research on text-to-video generation. Current mainstream video generation models are able to produce short clips with high quality (OpenAI, 2024; Kong

[1]Institute of Computing and Intelligence, Harbin Institute of Technology, Shenzhen, China [2]The University of Hong Kong. Correspondence to: Kehai Chen <chenkehai@hit.edu.cn>.

*Proceedings of the 43rd International Conference on Machine Learning*, Seoul, South Korea. PMLR 306, 2026. Copyright 2026 by the author(s).

*Table 1.* **Comparison of benchmarks in terms of sample scale, average length and complexity of their used prompts.** The implementation details are provided in Appendix B.1.

| Benchmarks | Samples | Avg. Words | Complexity |
|---|---|---|---|
| EvalCrafter (Liu et al., 2024b) | 700 | 12.33 | 3.74 |
| VBench-Long (Huang et al., 2025b) | 946 | 7.64 | 2.54 |
| VBench 2.0 (Zheng et al., 2025) | 90 | 125.46 | 8.13 |
| VMBench (Ling et al., 2025) | 1050 | 26.23 | 5.24 |
| AniMaker (Shi et al., 2025) | 50 | 143.34 | 5.75 |
| FilMaster (Huang et al., 2025a) | 10 | 95.70 | 8.07 |
| **LoCoT2V-Bench** *(ours)* | 234 | **248.85** | **8.70** |

et al., 2024; Runway AI, 2025; DeepMind, 2025; MiniMax, 2025; Kuaishou, 2025; Wan et al., 2025). However, they struggle to generate long-form and complex videos, which are inherently defined in this study by intrinsic video properties: durations exceeding 10 seconds, multiple scene transitions, and complex spatio-temporal dynamics. To overcome this limitation, some works optimize model architectures and training strategies for longer sequences (He et al., 2022; Lu et al., 2024; Henschel et al., 2025; Chen et al., 2025a), while others leverage Large Language Models (LLMs) to plan scripts and orchestrate multiple tools for multi-shot or story-level video creation (Long et al., 2024; Zhuang et al., 2024; Xie et al., 2024; Zheng et al., 2024). These efforts have pushed the frontier of long video generation forward.

As T2V models mature, the gap between casual curiosity and professional production becomes increasingly apparent. While casual users typically employ concise prompts and rely on a model's "random imagination," professional workflows (e.g., filmmaking and advertising) require deterministic control over intrinsic video properties. This control is achieved through detailed, script-level instructions with high semantic constraint density (e.g., precise character settings, choreographed camera movements, and multi-scene coherence) to strictly constrain the generation space (Hong et al., 2023; Zhou et al., 2024; Yang et al., 2026). Although recent advances have enabled more flexible approaches to long video generation, evaluating their performance under complex text inputs remains an open challenge. Researchers have made some progress in benchmarking video generation models (Huang et al., 2024; Liu et al., 2024b; Ling et al., 2025; Han et al., 2025; Zheng et al., 2025). These

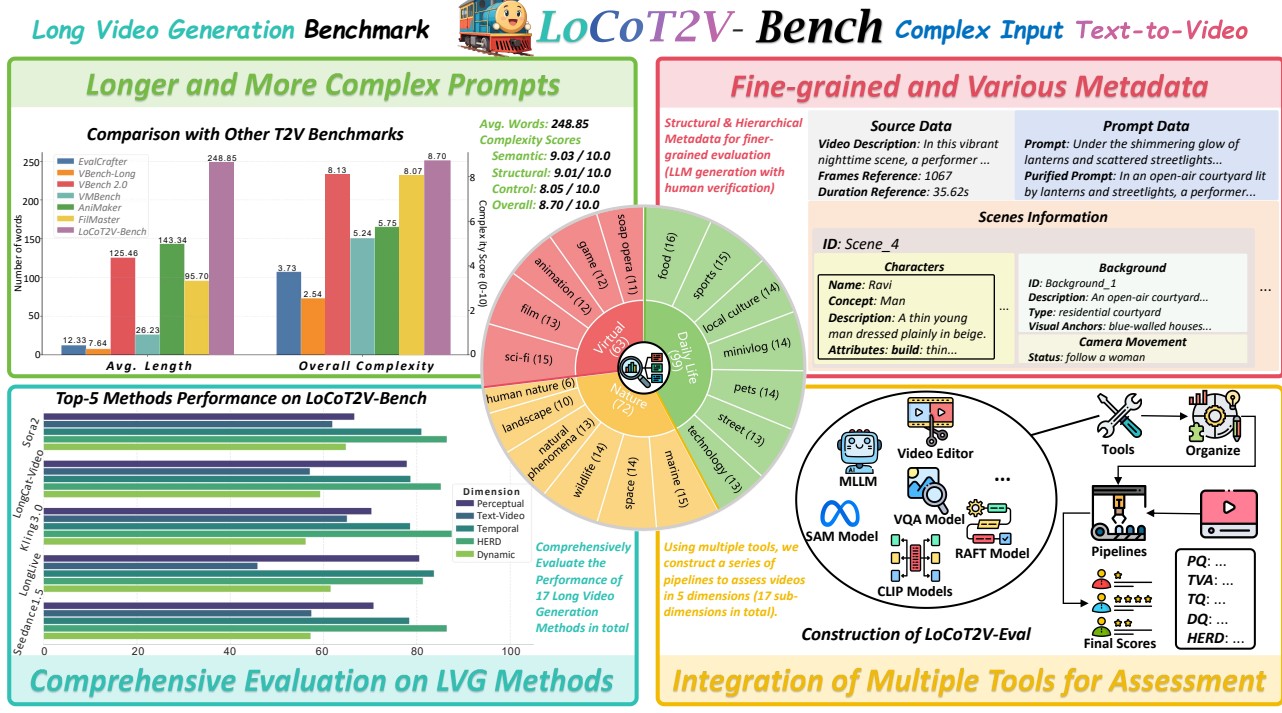

Figure 1. **Overview of LoCoT2V-Bench.** It's a benchmark for longer text-to-video generation with complex input. We construct structural and hierarchical metadata for our evaluation and leverage multiple tools to comprehensively assess the performance of 13 LVG methods.

works have proposed a comprehensive evaluation framework via delicate prompt construction and well-designed multi-dimensional metrics. However, most of them primarily target short videos. Their dependence on specific prompt construction strategies further limits their applicability to complex input scenarios, particularly when assessing longer videos in real-world settings. Moreover, existing benchmarks mainly emphasize visual quality, temporal consistency, and overall prompt adherence, while overlooking finer-grained alignment and higher-level aspects such as thematic expression. This limitation becomes even more pronounced when evaluating long-form videos.

To address these gaps, we introduce LoCoT2V-Bench, a benchmark for long-form text-to-video generation under dense semantic constraints and extended durations. Specifically, we collect 234 real-world videos spanning 18 themes. Instead of merely generating verbose descriptions via LLMs, we construct multi-scene prompts with hierarchical metadata (e.g., character settings, object dynamics, and camera behaviors) grounded in these real videos, bridging the gap between intentional scriptwriting and model execution. These prompts are longer and more complex than existing benchmarks as shown in Table 1. Building on this benchmark, we propose LoCoT2V-Eval, a multi-dimensional evaluation framework that jointly assesses perceptual quality, text-video alignment (overall and fine-grained), temporal quality, dynamic quality, and Human Expectation Realiza-

tion Degree (HERD) for narrative- and expectation-level attributes. Using LoCoT2V-Eval, we benchmark 17 representative LVG methods and provide diagnostic analyses of their strengths and limitations. We also validate LoCoT2V-Eval's alignment with human preferences through a series of analyses. **The main contributions are as follows:**

- **LoCoT2V-Bench:** a prompt suite constructed from 234 curated longer videos across 18 themes, featuring multi-scene prompts with hierarchical metadata.

- **LoCoT2V-Eval:** a multi-dimensional suite for LVG, covering perceptual quality, coarse-to-fine text-video alignment, temporal/dynamic quality, and the proposed HERD metric designed for higher-level dimensions.

- **Benchmarking and Analysis:** evaluation of 17 representative LVG methods reveals deficiencies in fine-grained text-video alignment and long-term character consistency. Further analyses show that LoCoT2V-Eval aligns well with human preferences.

## 2. Related Work

**Long Video Generation.** Long video generation requires models to generate videos longer than 10 seconds, and it has always been an essential direction in the field of video generation. Some of the existing methods are mainly based on diffusion models (He et al., 2022; Wang et al., 2023a;

Lu et al., 2024; Ouyang et al., 2024; Song et al., 2025; Henschel et al., 2025). These studies introduce carefully designed modules and training strategies for extending short video generation models to longer video generation. Other methods use autoregressive models for LVG (Ge et al., 2022; Villegas et al.; Chen et al., 2024; Weng et al., 2024; Yin et al., 2025; Chen et al., 2025a; Teng et al., 2025). Due to the autoregressive generation paradigm, they could support variable length and even ultra-long video generation. While these investigations are capable of generating long videos with high quality, most of them are limited to single-scene video generation, narrowing their application scope.

Another type of long video, multi-scene video, has also achieved significant advances in recent years (Lin et al.; Zhu et al., 2023; Long et al., 2024; Zhuang et al., 2024; Zheng et al., 2024). The considerable potential of LLM-driven agents towards tackling complex real-world problem has also brought some new thoughts into this area. For instance, Xie et al. (2024); Wu et al. (2025b) utilize powerful multi-agent collaboration to simulate film production procedures in reality, enabling more attractive and complex multi-scene video generation.

**Video Generation Evaluation.** The great progress of video generation triggers the development of benchmarks for evaluating these methods. Traditional approaches focus on frame-level image quality and diversity, such as Inception Score (IS) (Salimans et al., 2016), FID (Heusel et al., 2017), and FVD (Unterthiner et al., 2019). CLIP-Score (Hessel et al., 2021) is also used to evaluate prompt adherence of generated videos. However, given that video evaluation inherently involves multiple factors, these metrics remain limited in scope and lack more comprehensive assessment.

To fully evaluate the quality of the generated videos, a series of benchmarks have been proposed recently (Huang et al., 2024; Liu et al., 2024b; Ling et al., 2025; Qi et al., 2025; Yang et al., 2025b). These works design prompt suites and leverage various tools to construct multi-dimensional evaluation metrics. While such efforts have led to comprehensive evaluation frameworks for video generation, they typically employ prompts describing a single scene with limited content and mainly target short video evaluation. For long-form and complex text-to-video generation, Huang et al. (2025b) extends VBench (Huang et al., 2024) to support longer videos, and Zheng et al. (2025) further complement the evaluation suite with complex plot and landscape generation. However, the former still relies on simplified prompts, while the latter is constrained by the limited amount and complexity of its prompts. Bugliarello et al. (2023); Zhuang et al. (2025) assess videos from multi-prompt inputs, primarily focusing on story visualization rather than video generation.

## 3. LoCoT2V-Bench

LoCoT2V-Bench is constructed to rigorously evaluate video generation capabilities through high-quality, real-world data that aligns with professional production standards. In this section, we first introduce the collection and rigorous filtration process of source videos to ensure thematic diversity and visual fidelity (3.1). Then, we present the multi-stage pipeline for prompt suite construction (3.2), which leverages advanced MLLMs and LLMs for raw generation, narrative expansion, and meticulous post-processing.

### 3.1. Source Video Collection

To align our evaluation more closely with real-world video production, we collect thousands of short-form videos (30-60 seconds) from YouTube using yt-dlp[1] , guided by 18 predefined thematic keywords. We then manually filter out invalid samples, such as those occluded by subtitles/watermarks, exhibiting degraded visual quality, or showing misalignment between the content and the target theme. As illustrated in Fig. 1, the final dataset consists of 234 videos, evenly distributed across 18 themes.

### 3.2. Prompt Suite Construction

Recently an increasing number of MLLMs have demonstrated strong visual understanding capacities (Bai et al., 2025b; Zuo et al., 2025; Zhu et al., 2025b; Zhang et al., 2025a; Zhu et al., 2025a; Guo et al., 2025; Ma et al., 2026). Accordingly, we construct our prompt suite through a multi-stage MLLM-based generation pipeline combined with meticulous human verification as shown in Fig. 7.

**Raw Prompt Generation.** In this stage, we directly take the description text generated by Seed 1.5-VL (Guo et al., 2025) as the raw prompt for each collected video. To ensure the quality of these prompts, we carefully design the generation instructions and adopt the self-refine (Madaan et al., 2023) paradigm for iterative optimization.

**Prompt Content Expansion.** Considering that raw prompts are primarily descriptive texts, we use GPT-5 (OpenAI, 2025a) to expand them into story-like prompts with detailed character settings. We then manually refine the outputs to ensure: (1) **Rationality** (logical coherence without contradictions), (2) **Certainty** (avoiding uncertain expressions for unambiguous assessment), (3) **Character Completeness** (sufficiently distinctive attributes for reliable anchoring), and (4) **Consistency** (internally consistent traits and actions). Finally, we employ GPT-5 to perform an additional round of automatic checking over all prompts, followed by manual re-inspection of potentially problematic samples to ensure that the prompts can meet our expectations.

---

[1]https://github.com/yt-dlp/yt-dlp

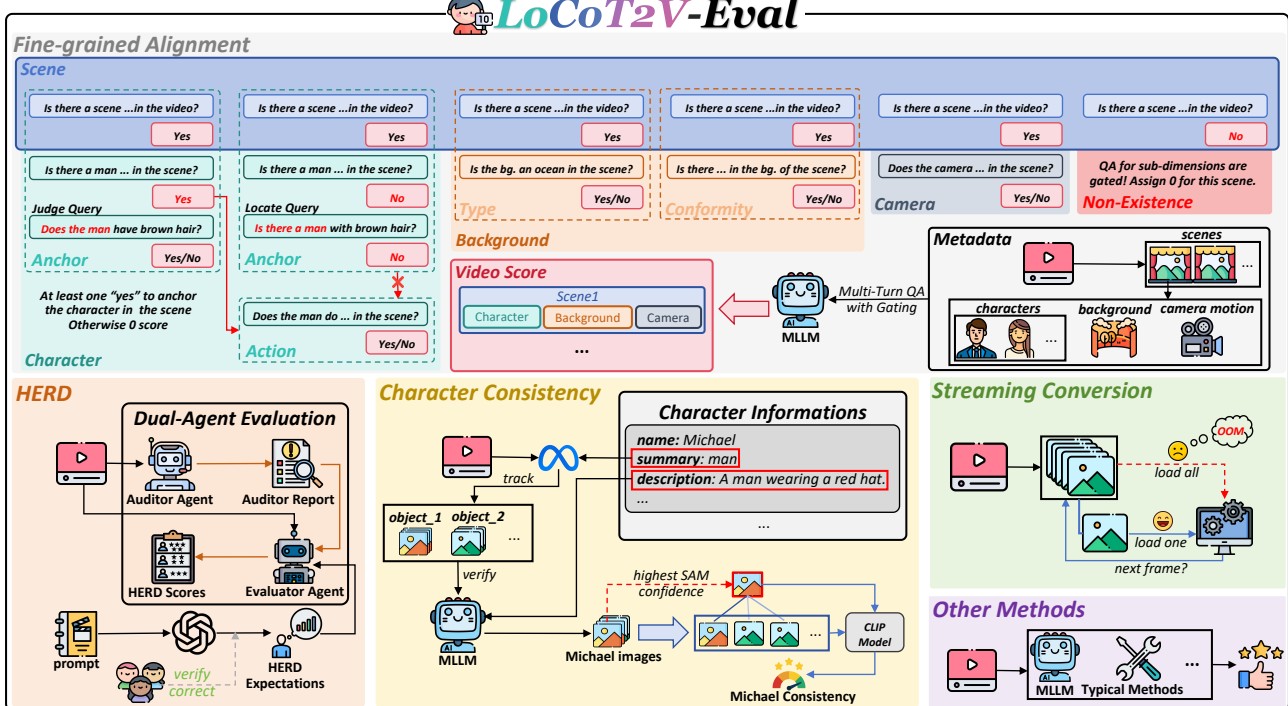

*Figure 2.* **Overview of LoCoT2V-Eval.** We demonstrate key parts of our proposed framework LoCoT2V-Eval: Fine-grained Alignment, Character Consistency, and HERD. Some existing implementations are converted into a streaming style to avoid memory overwhelming.

**Prompt Post-processing.** We observe that LLMs like GPT-5 tend to repeatedly assign certain character names when introducing characters into prompts, as illustrated in Fig. 7 in Appendix B.2. To mitigate this repetition and enhance the name diversity, we track the occurrence of each character name and leverage GPT-5 to generate a new name whenever a duplicate is detected. In addition, we identify and refine potentially unsafe or sensitive content, such as depictions of blood or violence, with the assistance of GPT-5. Human verification is also incorporated throughout the post-processing steps to ensure correctness and consistency. After these procedures, the prompts will be used for evaluation.

## 4. LoCoT2V-Eval

To comprehensively and multidimensionally evaluate the quality of videos generated by different T2V models via LoCoT2V-Bench, we propose the LoCoT2V-Eval framework. In this section, we first introduce Perceptual Quality to quantify visual fidelity (4.1) and Text-Video Alignment to assess semantic faithfulness through a coarse-to-fine strategy (4.2). Subsequently, addressing the intrinsic temporal nature of video content, we detail Temporal Quality for evaluating consistency across frames (4.3) and Dynamic Quality for measuring motion richness and smoothness (4.4). Finally, to capture high-level narrative and emotional nuances, we present the Human Expectations Realization Degree

(HERD), a novel metric to bridge the gap between objective analysis and subjective human perception (4.5).

### 4.1. Perceptual Quality (PQ)

Perceptual quality is defined as the overall visual quality of a video as perceived by human observers, reflecting both aesthetic appeal and technical fidelity. This metric primarily focuses on frame-level image quality. Motivated by recent advances in Image Quality Assessment (IQA) (Wang et al., 2023b; Wu et al., 2024; Zhu et al., 2024; Li et al., 2026; You et al., 2025), we compute perceptual quality scores for each video by combining DeQA-Score (You et al., 2025) with a multi-scale pyramid frame sampling strategy.

Specifically, for each temporal scale with window ratio $\alpha$, the video is partitioned into non-overlapping windows of length $L = \lfloor N \cdot \alpha \rfloor$. Within each window, $n_\alpha = \lfloor L \cdot \beta_\alpha \rfloor$ frames are sampled symmetrically around the center, $\beta_\alpha$ refers to the sampling ratio corresponding to $\alpha$. The frame-level quality scores are then computed using DeQA-Score, averaged within each window, and finally averaged across all windows to yield the video-level perceptual quality score:

$$\text{PQ}(v) = \frac{1}{|W|} \sum_{w \in W} \frac{1}{n_\alpha} \sum_{f \in w} \text{DeQA}(f),$$

where $v$ denotes the video, $W$ is the set of temporal windows, $f$ indexes the sampled frames in each window, and

DeQA($\cdot$) is the DeQA-Score method producing the image quality score for each frame. This procedure is applied independently at multiple temporal scales to extract various images and capture both local and global perceptual quality.

### 4.2. Text-Video Alignment (TVA)

Text-video alignment measures the adherence and faithfulness of the video content to the input text prompt. Considering the complexity of prompts, we evaluate them in a coarse-to-fine manner, including overall alignment and fine-grained alignment.

**Overall Alignment (OA).** Overall alignment evaluates the global consistency between a video and its corresponding text prompt. Existing benchmarks predominantly rely on CLIP-based methods (Huang et al., 2024; Liu et al., 2024b; Qi et al., 2025; Wu et al., 2025a). However, as CLIP was originally designed for image-text representation learning (Radford et al., 2021), such approaches are inherently limited in capturing complex video-text relationships, especially for complex prompts.

To overcome this limitation, we leverage Qwen3-VL-8B (Bai et al., 2025a) with great video understanding capabilities to assess overall alignment by assigning a score to each video based on how well it matches its prompt across multiple aspects, including characters, settings, and interactions. Scores are reported on a 100-point scale rather than a commonly used 5-point scale, enabling finer-grained differentiation during evaluation.

**Fine-grained Alignment (FGA).** The intricate nature of complex prompts calls for a fine-grained evaluation mechanism to assess the fidelity of generated videos. Inspired by VQA-based methods (Hu et al., 2024; Jing et al.), we introduce a hierarchical, tree-structured VQA framework that decomposes the alignment into conditional, scene-wise verification steps (see Fig. 2). Given an input prompt, we parse it into a sequence of scenes $\mathcal{S} = \{s_1, \ldots, s_{|\mathcal{S}|}\}$ and extract structured metadata along three axes: *Character*, *Background*, and *Camera Movement*. To formally represent the multi-turn dialogue context, let $\mathcal{H}$ denote the accumulating history of Question-Answer pairs. The VQA model's evaluation of a query $q$ is strictly conditioned on this context, denoted as $\mathcal{A}(q|\mathcal{H}) \in \{0, 1\}$, where 1 corresponds to an affirmed response ("Yes") and 0 to a negative one ("No").

The evaluation begins at the scene level. For each scene $s$, an initial existence query $q_s^{\text{exist}}$ defines a scene-level gate $g(s) = \mathcal{A}(q_s^{\text{exist}}|\varnothing)$. If $g(s) = 0$, the entire evaluation subtree for scene $s$ is pruned. Otherwise, the affirmed QA pair is retained to initialize the scene context $\mathcal{H}_s = \{(q_s^{\text{exist}}, 1)\}$, thereby anchoring all descendant queries to this confirmed temporal segment. For a valid scene ($g(s) = 1$), each applicable facet $b \in \mathcal{B}_s \subseteq \{\text{char}, \text{bg}, \text{cam}\}$ is evaluated

independently under $\mathcal{H}_s$.

The *Character* facet involves assessing $N_c$ static attributes and $M_c$ dynamic actions for each character $c \in \mathcal{C}_s$. To strictly align with the multi-turn logic in Fig. 2, we propose a **state-aware adaptive anchoring** mechanism. We maintain a dynamic anchoring flag $a_s^c \in \{0, 1\}$ (initially 0) and an evolving context $\mathcal{H}_{k-1}$ (starting from $\mathcal{H}_s$). At step $k \in \{1, \ldots, N_c\}$, the query $q_{c,k}$ is formulated based on the current $a_s^c$: it is a *Locate Query* (e.g., "*Is there a tall man...*") if $a_s^c = 0$, and switches to a grounded *Judge Query* (e.g., "*Is the man tall?*") once $a_s^c = 1$. The response is evaluated as $y_k = \mathcal{A}(q_{c,k}|\mathcal{H}_{k-1})$. If a locate query is affirmed ($y_k = 1$ when $a_s^c = 0$), the character is successfully anchored, instantly toggling $a_s^c \leftarrow 1$. The dialogue context is explicitly inherited and updated step-by-step: $\mathcal{H}_k = \mathcal{H}_{k-1} \cup \{(q_{c,k}, y_k)\}$. Thus, the attribute score naturally emerges as:

$$f_{\text{attr}}^c = \frac{1}{N_c} \sum_{k=1}^{N_c} y_k. \tag{1}$$

Following the attribute verification loop, the *Action* verification is strictly gated by the final anchoring status $a_s^c$. Let $Q_{s,c}^{\text{act}}$ denote the set of $M_c$ parallel action queries for character $c$. These queries are evaluated conditioned on the fully accumulated attribute context $\mathcal{H}_{N_c}$. We explicitly formulate the gated action score using $a_s^c$ as a multiplicative factor:

$$f_{\text{action}}^c = a_s^c \cdot \frac{1}{M_c} \sum_{q \in Q_{s,c}^{\text{act}}} \mathcal{A}(q \mid \mathcal{H}_{N_c}). \tag{2}$$

Consequently, if the character is never anchored ($a_s^c = 0$), the action evaluation evaluates to zero, preventing hallucinated action scores. The character score for scene $s$ is derived by averaging the combined performance across all entities:

$$f_s^{\text{char}} = \frac{1}{|\mathcal{C}_s|} \sum_{c \in \mathcal{C}_s} \frac{1}{2}(f_{\text{attr}}^c + f_{\text{action}}^c). \tag{3}$$

The remaining facets are evaluated straightforwardly, conditioned on the initialized scene context $\mathcal{H}_s$. For the *Background* facet, the evaluation comprises a single type query $q_s^{\text{bg\_type}}$ to verify the global environmental setting, and a set of conformity queries $Q_s^{\text{bg\_conf}}$ to check for specific visual anchors. We define the background score $f_s^{\text{bg}}$ as the arithmetic mean of the type verification and the average conformity performance:

$$f_s^{\text{bg}} = \frac{1}{2}\left(\mathcal{A}(q_s^{\text{bg\_type}} \mid \mathcal{H}_s) + \frac{\sum_{q \in Q_s^{\text{bg\_conf}}} \mathcal{A}(q \mid \mathcal{H}_s)}{|Q_s^{\text{bg\_conf}}|}\right). \tag{4}$$

Similarly, the *Camera Movement* facet relies on a single query $q_s^{\text{cam}}$ to verify the prescribed cinematic motion (e.g.,

*pan*, *zoom*), with its score simply being $f_s^{\text{cam}} = \mathcal{A}(q_s^{\text{cam}} \mid \mathcal{H}_s)$. Ultimately, for each valid scene, the score is

$$F(s) = \frac{1}{|\mathcal{B}_s|} \sum_{b \in \mathcal{B}_s} f_s^b, \qquad (5)$$

and the final Fine-Grained Alignment (FGA) score aggregates these scene-level results:

$$S_{\text{FGA}} = \frac{1}{|\mathcal{S}|} \sum_{s \in \mathcal{S}} g(s) \cdot F(s). \qquad (6)$$

By factoring the gate $g(s)$ into the numerator while retaining the original scene count $|\mathcal{S}|$ in the denominator, we explicitly penalizes the model for hallucinated or missing scenes.

### 4.3. Temporal Quality (TQ)

Temporal quality primarily measures the consistency and coherence of video frames. To comprehensively evaluate this dimension, we split it into the following aspects and design corresponding evaluation methods.

**Character Consistency (CC).** This evaluates the temporal stability of character identity. To assess it, we propose a multi-stage pipeline as shown in Fig 2. First, we employ SAM3 (Carion et al., 2025) to extract character trajectories. Leveraging the structured metadata built in 4.2, we construct attribute-augmented short prompt for each character (e.g., *woman in white*) to guide the SAM3. To further mitigate tracking errors, a strong MLLM acts as a verifier, retaining only those instances that align with the character's description (In our practice, we use Qwen3-VL-8B as the verifier out of the trade-off between performance and deployment difficulty). Finally, we compute the consistency score as the average cosine similarity between the visual embeddings via FG-CLIP2 (Xie et al., 2025a) of the highest-confidence anchor instance and all other verified instances in the video.

**Background Consistency (BC).** Unlike character consistency, background consistency emphasizes the stability of the environment across continuous shots. Some prior work removes characters via segmentation and computes semantic similarity over the remaining regions across consecutive frames (Fei et al., 2025). However, since CLIP-based models (Radford et al., 2021; Zhai et al., 2023; Tschannen et al., 2025; Xie et al., 2025a) are primarily trained on complete images, we instead follow the paradigm adopted in VBench (Huang et al., 2024). Specifically, we use the same CLIP model, FG-CLIP2 (Xie et al., 2025a), as in character consistency to compute semantic similarity between adjacent frames, and implement the metric in a streaming manner to accommodate long videos.

**Warping Error (WE).** Warping error measures pixel-level discrepancies between consecutive frames after optical-flow-based alignment, revealing temporal artifacts such as flickering. Following EvalCrafter (Liu et al., 2024b), we compute

warping error and further modify its evaluation protocol into a streaming version to accommodate long-video evaluation. The resulting scores are mapped to $[0, 1]$ using exponential normalization $e^{-ax}$ for better interpretability.

### 4.4. Dynamic Quality (DQ)

Dynamic quality evaluates the extent of temporal changes and the smoothness of these changes, encompassing both motion dynamics and content-level variation. This metric aims to capture whether a video exhibits sufficient and continuous temporal progression, rather than remaining visually static or repetitive over time. To measure it at different time scales, we evaluate this metric at three granularities: frame-level, segment-level, and video-level. The overall score is obtained by applying a pre-fitted linear model to aggregate the sub-dimension scores. Please refer to Appendix B.3 for more implementation details.

**Frame-level Dynamic Quality.** Frame-level dynamic quality focuses on the temporal dynamics between consecutive frames. We adopt two metrics from VBench (Huang et al., 2024): **Dynamic Degree**, which evaluates whether a video exhibits sufficient motion to reflect its overall level of dynamics, and **Motion Smoothness**, which assesses whether the motion is smooth and physically consistent with real-world dynamics. Both are slightly modified to better suit our evaluation setting and are also adapted to a streaming version for long videos.

**High-level Dynamic Quality.** High-level dynamic quality aims to capture long-range temporal dynamics across video segments and the entire sequence—an aspect largely overlooked by existing benchmarks (Huang et al., 2024; Liu et al., 2024b; Ling et al., 2025), but partially addressed in DEVIL (Liao et al., 2024). Building upon DEVIL, we adapt its metrics to a streaming setting to support long videos. Specifically, we evaluate DQ at both the segment and video levels: segment-level DQ measures fine-grained and global aperiodicity to reflect dynamic diversity across segments, while video-level DQ characterizes high-level temporal variation and low-level information flow over the full sequence.

### 4.5. Human Expectations Realization Degree (HERD)

Current video generation benchmarks struggle to assess high-level aspects such as emotional response and thematic expression. To address this limitation, we propose the Human Expectation Realization Degree (HERD), a metric designed to evaluate how well a generated video fulfills human expectations inferred from a text prompt. HERD comprises six dimensions: *Emotional Response*, *Narrative Flow*, *Character Development*, *Visual Style*, *Themes Expression*, and *Overall Impression* (see Appendix B.4 for their definitions).

Given a prompt, we first employ GPT-5 (OpenAI, 2025a) to

*Table 2.* **Performance results of all baseline methods evaluated over the five major dimensions.** Scores for each dimension are obtained by averaging the scores of its corresponding sub-dimensions. Sub-dimension scores of dynamic quality and HERD are presented in Appendix C.5. Note that all values are expressed as percentages to improve readability and conserve space.

| Method | Perceptual Quality | Text-Video Alignment | | | Temporal Quality | | | | HERD | Dynamic Quality | Avg. |
|---|---|---|---|---|---|---|---|---|---|---|---|
| | | OA | FGA | Avg. | CC | BC | WE | Avg. | | | |
| *Multi-prompt Input Methods* | | | | | | | | | | | |
| FreeNoise (Qiu et al., 2024) | 73.89 | 18.12 | 10.38 | 14.25 | 15.38 | 98.77 | 95.39 | 69.85 | 53.65 | 50.55 | 52.44 |
| MEVG (Oh et al., 2024) | 40.74 | 23.08 | 10.81 | 16.95 | 5.55 | 90.68 | 90.05 | 62.09 | 44.93 | 33.78 | 39.70 |
| FreeLong (Lu et al., 2024) | 66.78 | 35.26 | 16.10 | 25.68 | 20.98 | 97.52 | 92.05 | 70.18 | 58.03 | 46.10 | 53.35 |
| DiTCtrl (Cai et al., 2025) | 56.55 | 48.25 | 45.54 | 46.90 | 25.72 | 96.86 | 94.92 | 72.50 | 60.75 | 49.37 | 57.21 |
| StoryAdapter (Mao et al., 2024) | 79.62 | 52.89 | 36.00 | 44.45 | 39.05 | 98.90 | 95.11 | 77.69 | 68.76 | 50.36 | 64.17 |
| *Direct Input Methods* | | | | | | | | | | | |
| CausVid (Yin et al., 2025) | 69.23 | 44.96 | 26.73 | 35.85 | 45.97 | 98.98 | 95.76 | 80.24 | 68.53 | 58.06 | 62.38 |
| FIFO-Diffusion (Kim et al., 2024) | 70.94 | 13.89 | 9.94 | 11.92 | 20.26 | 98.08 | 89.95 | 69.43 | 66.87 | 53.72 | 54.58 |
| SkyReels-V2 (Chen et al., 2025a) | 70.43 | 52.80 | 38.25 | 45.53 | 29.19 | 98.97 | 95.52 | 74.56 | 79.56 | 60.24 | 66.06 |
| Vlogger (Zhuang et al., 2024) | 62.96 | 37.29 | 29.47 | 33.38 | 23.11 | 95.64 | 85.08 | 67.94 | 60.23 | 41.37 | 53.18 |
| VGoT (Zheng et al., 2024) | 82.73 | 49.64 | 29.71 | 39.68 | 32.66 | 99.44 | 97.59 | 76.56 | 67.89 | 59.83 | 65.34 |
| SANA-Video (Chen et al., 2025b) | 73.99 | 41.75 | 26.19 | 33.97 | 28.17 | 98.52 | 90.89 | 72.53 | 70.68 | 59.30 | 62.09 |
| LongLive (Yang et al., 2025a) | 80.51 | 55.50 | 36.15 | 45.83 | **54.92** | 99.18 | 96.89 | **83.66** | 81.30 | 61.52 | 70.56 |
| LongSANA (Chen et al., 2025b) | **84.11** | 38.06 | 21.52 | 29.79 | 39.28 | **99.53** | 97.61 | 78.81 | 78.70 | 59.73 | 66.23 |
| LongCat-Video (Team et al., 2025) | 77.75 | 65.59 | 51.01 | 58.30 | 42.08 | 98.31 | 94.95 | 78.45 | 84.80 | 59.29 | 71.72 |
| Sora2 (OpenAI, 2025b) | 66.59 | 69.64 | 54.09 | 61.87 | 45.40 | 99.10 | 98.41 | 80.97 | 86.42 | **64.78** | **72.13** |
| Seedance 1.5-Pro (Seedance et al., 2025) | 70.71 | 67.58 | 47.17 | 57.38 | 38.25 | 99.13 | 97.72 | 78.37 | 86.41 | 57.21 | 70.02 |
| Kling 3.0 (Kuaishou, 2025) | 70.26 | **73.08** | **56.94** | **65.01** | 36.97 | 98.96 | **99.73** | 78.55 | **87.47** | 56.16 | 71.49 |

generate structured *HERD expectations* as a prompt-centric reference. To evaluate expectation realization while reducing subjective bias, we adopt a decoupled *Auditor-Evaluator* framework (Fig. 2). The Auditor analyzes the generated video without access to the expectations and produces a factual report on visual content, temporal stability, and artifacts (e.g., flickering). The Evaluator is then provided with both the report and the video to mitigate potential hallucinations in auditor-generated content, and assigns scores from 1 to 5 for each dimension based on expectation fulfillment. Finally we compute the macro-average of the normalized dimension scores $s_d \in [0, 1]$ to obtain the HERD score for each video, defined as:

$$S_{\text{HERD}} = \frac{1}{|\mathcal{D}|} \sum_{d \in \mathcal{D}} s_d, \qquad (7)$$

where $\mathcal{D}$ denotes the set of six above-mentioned dimensions.

## 5. Experiments

In our experiments, we select 17 representative long-form video generation (LVG) methods as baselines and evaluate their generated videos using LoCoT2V-Eval. The results are organized into two splits according to the input prompt format. Specifically, *multi-prompt input* refers to methods that require the original prompt to be decomposed into scene-level prompts, whereas *direct input* denotes methods that directly take the original prompt as input without prompt decomposition. Detailed descriptions and implementations of these methods are provided in Appendix B.5.

### 5.1. Main Results

As presented in Table 2, existing LVG methods demonstrate robust capabilities in frame-level *Perceptual Quality*, with models like LongSANA reaching 84.11%. They also achieve exceptional stability in environment generation, evidenced by *Background Consistency (BC)* and *Warping Error (WE)* scores consistently exceeding 90% across most baselines. However, a significant performance gap exists in finer-grained and temporal dimensions. Specifically, there is a sharp contrast between the high BC scores and the substantially lower *Character Consistency (CC)*, where most methods score below 50%, highlighting the difficulty of maintaining subject identity compared to static backgrounds. Similarly, in terms of text–video alignment, the notable drop from *Overall Alignment (OA)* to *Fine-Grained Alignment (FGA)* reveals limitations in handling the intricacies of complex test prompts. Among the evaluated approaches, advanced proprietary models like Kling 3.0 and Sora2 exhibit superior overall performance, particularly leading in *HERD* (up to 87.47%) and text-video alignment, suggesting a better alignment with human expectations compared with most of the open-source methods, while LongLive could still stand out with its highest *Temporal Quality* score (83.66%). Nevertheless, despite the advancements brought by recent leading models, the relatively low scores in FGA and CC underscore that generating long, semantically precise, and temporally consistent videos remains an open challenge.

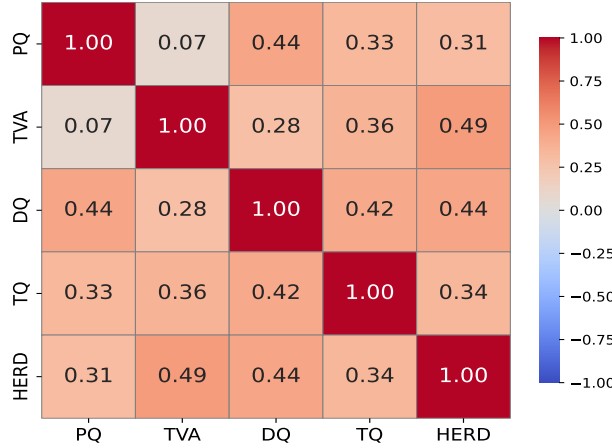

*Figure 3.* **Correlations between different evaluation dimensions.** Correlations are computed across all generated samples.

### 5.2. Analysis

**Correlations between Different Metrics.** To analyze the relationships among different evaluation dimensions, we compute the Spearman rank correlations across all generated videos. As illustrated in Fig. 3, non-negligible correlations are observed among these dimensions. Two notable patterns emerge. First, TVA exhibits stronger correlations with TQ and HERD than the remaining dimensions, likely because all three depend more heavily on prompt-related semantics. Second, aside from TVA, PQ shows consistently strong correlations with DQ, TQ, and HERD, indicating that videos with higher visual quality tend to achieve better overall evaluation scores. This observation aligns with the general expectation that visually higher-quality outputs are typically associated with stronger generation performance.

*Table 3.* **Human alignment results for multiple dimensions.** All results are expressed as percentages to improve readability.

| Metrics | Sub-Dimensions | PLCC | SRCC | KRCC |
|---|---|---|---|---|
| PQ | Perceptual Quality | **71.39** | **70.20** | **54.67** |
| TVA | Overall Alignment | 63.38 | 67.23 | 52.68 |
| | Fine-grained Alignment | 53.55 | 52.70 | 37.96 |
| TQ | Character Consistency | 47.17 | 49.90 | 39.19 |
| | Background Consistency | 52.80 | 51.11 | 36.57 |
| | *w/ character removal* | 18.63 | 30.98 | 21.59 |
| HERD | Emotional Response | 52.23 | 48.58 | 40.08 |
| | Narrative Flow | 48.78 | 48.24 | 39.33 |
| | Character Development | 42.70 | 41.06 | 32.27 |
| | Visual Style | 43.27 | 44.08 | 35.47 |
| | Themes Expression | 45.54 | 42.48 | 34.90 |
| | Overall Impression | 46.65 | 43.06 | 35.27 |
| | HERD Score | 54.86 | 58.44 | 44.92 |
| DQ | Dynamic Quality | 53.86 | 54.81 | 41.01 |

**Human Alignment Analysis.** Human alignment is a criti-

cal requirement for reliable video generation evaluation. To assess the alignment of LoCoT2V-Eval with human judgment, we randomly sampled 5% of the generated videos and asked three experienced annotators to score them along the five predefined dimensions (see Appendix B.6 for details). As shown in Table 3, we report the Spearman Rank Correlation Coefficient (SRCC), Pearson Linear Correlation Coefficient (PLCC), and Kendall's Rank Correlation Coefficient (KRCC) between the automatic evaluation scores and human annotations across nearly all sub-dimensions. Metrics closely related to visual quality and prompt adherence, such as perceptual quality and text-to-video alignment, demonstrate strong correlations with human preferences, while other aspects, including dynamic quality, exhibit moderate yet consistent alignment. Furthermore, the proposed HERD evaluation method achieves favorable correlation scores not only for the overall evaluation but also across its sub-dimensions. These results confirm the effectiveness of LoCoT2V-Eval in aligning with human judgment.

**Effect of Character Removal for BC.** As mentioned in 4.3, we compute frame-to-frame similarities for Background Consistency (BC) using complete images rather than masking character regions. To investigate which approach better aligns with human preferences, we also evaluate BC using frames with character regions masked. Specifically, we leverage the character information used in the character consistency evaluation and apply SAM3 (Carion et al., 2025) to extract all possible character masks. Each frame is then masked accordingly, and similarities are computed over the resulting images to obtain an alternative BC score. The alignment results are reported in Table 3, which shows that this character-removal approach performs substantially worse than using the complete frames, further supporting the validity of our chosen method.

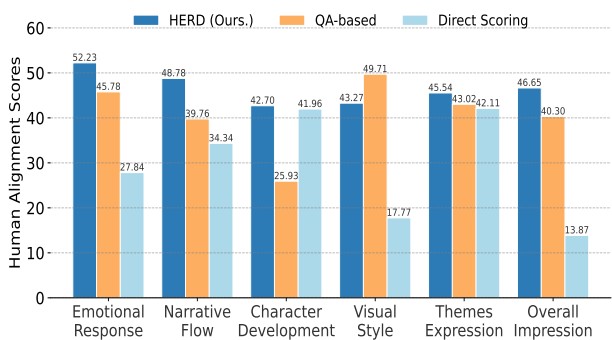

*Figure 4.* **Human alignment comparison among different HERD implementations.** We display PLCC scores in this figure.

**Comparison of Evaluation Methods for HERD.** As we adopt a dual-agent framework to compute HERD scores for each video in 4.5, we also evaluate several alternative strategies including QA-based evaluation and directly prompting Qwen3-VL-8B (Bai et al., 2025a) to score each HERD sub-

*Table 4.* **Human Alignment Results on Shared Dimensions with VBench-Long.** We employ the same scores as those in Table 3.

| Benchmark | Perceptual Quality | | | Overall Alignment | | | Character Consistency | | | Background Consistency | | |
|---|---|---|---|---|---|---|---|---|---|---|---|---|
| | PLCC | SRCC | KRCC | PLCC | SRCC | KRCC | PLCC | SRCC | KRCC | PLCC | SRCC | KRCC |
| VBench-Long (Huang et al., 2025b) | 55.10 | 50.69 | 36.77 | 10.99 | 18.18 | 12.04 | 41.09 | 32.65 | 22.26 | 45.06 | 35.38 | 17.53 |
| **LoCoT2V-Bench (ours.)** | **71.39** | **70.20** | **54.67** | **63.38** | **67.23** | **52.68** | **47.17** | **49.90** | **39.19** | **52.80** | **51.11** | **36.57** |

dimension, and compare their alignment with human judgments (see Appendix B.7 for details). As shown in Fig. 4, our method consistently achieves higher and more stable human-alignment across nearly all HERD sub-dimensions. One notable exception is the *Visual Style* dimension, where the QA-based method exhibits slightly stronger alignment. We conjecture that this stems from its stronger emphasis on visual cues, whereas our additional textual auditor reports may introduce hallucinations during assessment, potentially grounded in the modality preference observed in (Zhang et al., 2025b).

### 5.3. Discussion

**Differentiation with VBench-Long.** In text-to-video area, VBench-Long (Huang et al., 2025b) is a widely adopted benchmark for LVG evaluation. However, it primarily targets isolated attributes via short, single-scene prompts, while our LoCoT2V-Bench evaluates professional narratives involving complex multi-scene interactions and fine-grained visual details (cases of our prompts could be seen in Appendix E). We further observe that several evaluation dimensions in VBench-Long overlap with those considered in our benchmark. To quantitatively compare different evaluation methods on these shared dimensions, we measure the correlation between VBench-Long's evaluation results and human judgments following the same protocol used in our previous experiments, and compare them with the results reported in Table. 3. The comparison is also presented in Table 4. The results show that our evaluation method achieves stronger alignment with human assessment on the shared dimensions. Given the broad adoption and recognition of VBench-Long, this comparison further demonstrates the effectiveness of our proposed evaluation framework.

**Temporal Complexity of Our Prompts.** While we provide a detailed definition of prompt complexity in Appendix B.1, benchmarks like StoryEval-Bench (Wang et al., 2025) emphasize *temporal complexity*, i.e., the complexity of narrative progression, state transitions, and causal dependencies beyond simple camera motion. To verify that our prompts also exhibit this property, we construct an evaluation prompt (see the prompt in Appendix D.5) and employ GPT-5 (OpenAI, 2025a) as the evaluator. We additionally report the results of StoryEval-Bench as a reference to support the validity of our evaluation methodology. Results in Fig. 5 indicate that: (1) prompts from StoryEval-Bench consistently

exhibit the key elements associated with temporal complexity, supporting the validity of our evaluation methodology; (2) although not all prompts in our benchmark strictly follow the temporal complexity definition adopted by StoryEval-Bench, LoCoT2V-Bench still achieves higher overall scores while containing more events on average, further demonstrating the temporal complexity of our prompts.

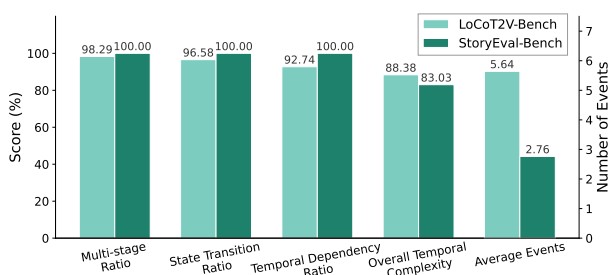

*Figure 5.* **Comparison Results with StoryEval-Bench in Temporal Complexity.** "Ratio" refers to the portion of prompts that have the corresponding elements related to specific sub-dimensions.

**Effect of the Prompt Categories.** As described in Section 3.1, we divide our prompts into three categories. We further report the scores across the five major evaluation dimensions for samples from different categories in Table 5 of Appendix C.3. These results serve as complementary evidence for the statistical reliability of our LoCoT2V-Eval. We also observe several notable trends. For example, the evaluated methods generally achieve substantially higher overall scores on "Daily Life" samples than on "Virtual" samples, and the best model varies across prompt categories.

## 6. Conclusion

We present LoCoT2V-Bench, a benchmark for evaluating long-form and complex text-to-video generation, along with a multi-dimensional and holistic evaluation framework, LoCoT2V-Eval. Experiments on 17 representative methods reveal that while current approaches achieve strong perceptual quality and global temporal stability, they struggle with fine-grained text-video alignment and local temporal coherence. Further relevant analyses demonstrate the robustness and diagnostic value of LoCoT2V-Bench. We hope this benchmark will facilitate more rigorous evaluation and inspire future research toward coherent, controllable, and human-aligned long-form video generation.

## Acknowledgements

This work was supported in part by the Science Fund for Creative Research Groups of the National Natural Science Foundation of China under Grant 62521006, in part by the National Natural Science Foundation of China (62276077, U23B2055, 62350710797), in part by Guangdong S&T Program (2024B0101050003), in part by the Guangdong Basic and Applied Basic Research Foundation (2024A1515011205), and in part by Shenzhen Science and Technology Program (KQTD20240729102154066).

## Impact Statement

All video data in LoCoT2V-Bench are collected from YouTube using yt-dlp in compliance with the platform's terms of service and copyright regulations. A rigorous filtering process, combining automatic checks and manual review, was applied to exclude invalid or harmful content. Prompts were generated and refined using VLMs and LLMs under strict instructions prohibiting PII, offensive, violent, or otherwise inappropriate material, with additional human verification to ensure factual accuracy and ethical compliance. During our evaluation, no private or sensitive data were used, and all procedures adhered to relevant ethical guidelines for AIGC research, ensuring LoCoT2V-Bench promotes safe and responsible development of long-form text-to-video generation technology.

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

# A. List of the Appendix Content

The content of our appendix is organized as follows:

# B. Implementation Details & Explanation

## B.1. Complexity Comparison

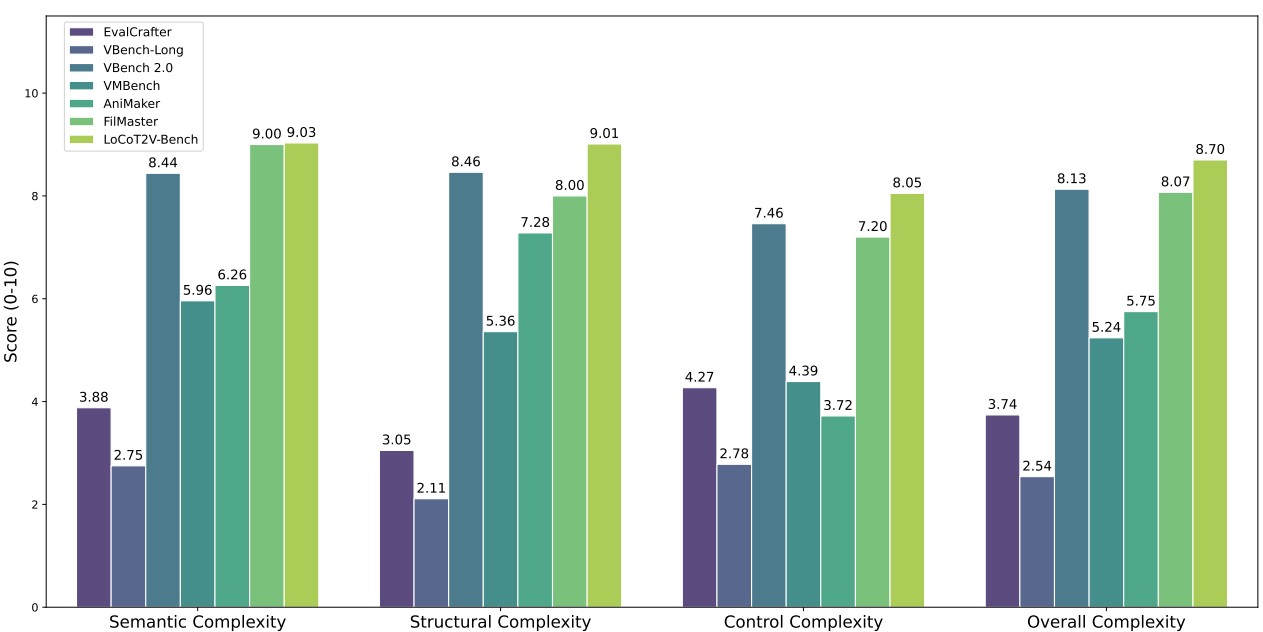

*Figure 6.* **Detailed Complexity Comparison among Different Benchmarks.** Scores of all types of defined complexity are provided.

As shown in Table 1, we report the complexity scores of prompts used in each benchmark. Specifically, the complexity of each prompt is directly assessed using DeepSeek-V3.2 (Liu et al., 2024a) according to predefined criteria along three dimensions. The overall complexity score is computed as the average of the scores across these dimensions, and the comparative results are visualized in Fig. 6. The prompt template employed for complexity scoring is provided in Appendix D.1. Below, we briefly introduce the evaluated benchmarks and describe how their prompts are incorporated into our complexity analysis:

- **EvalCrafter** (Liu et al., 2024b) is a comprehensive evaluation framework for text-to-video (T2V) models. It constructs a diverse benchmark consisting of 700 prompts, derived from real-world user data and further refined with the assistance of a large language model to ensure broad coverage. We directly adopt these prompts for complexity scoring.

- **VBench-Long** (Huang et al., 2025b) is an extension of VBench (Huang et al., 2024) that aims to address the misalignment between existing automatic metrics and human perceptual judgments. While extending the evaluation to longer videos, VBench-Long retains the original prompt set, which we use unchanged for complexity computation.

- **VBench 2.0** (Zheng et al., 2025) is a complementary version of VBench (Huang et al., 2024) that advances beyond surface-level video quality assessment to evaluate whether generated videos conform to fundamental real-world principles. To support more challenging input scenarios, VBench 2.0 introduces two subsets of prompts: *complex plot* (60 prompts) and *complex landscape* (30 prompts). The former emphasizes richer plot structures, whereas the latter focuses on long-form landscape depiction. Given their greater difficulty, we select these two subsets for complexity comparison.

- **VMBench** (Ling et al., 2025) presents the first comprehensive benchmark dedicated to evaluating video motion quality from a human perception perspective. It provides a diverse collection of motion-centric prompts spanning six fundamental dynamic scene dimensions: fluid dynamics, biological motion, mechanical motion, weather phenomena, collective behavior, and energy transfer. All prompts in VMBench are utilized for our complexity analysis.

- **AniMaker** (Shi et al., 2025) proposes a multi-agent framework for automatically generating coherent, long-form animated stories from textual narratives. Its evaluation benchmark is constructed from the TinyStories dataset and features prompts involving complex multi-character interactions across diverse settings. We use these prompts to assess prompt complexity.

- **FilMaster** (Huang et al., 2025a) is an end-to-end AI-driven film generation system that integrates real-world filmmaking principles to bridge the gap between generative models and professional cinematic standards. It introduces a holistic benchmark, FilmEval, which evaluates AI-generated films along six high-level dimensions. FilmEval includes 20 test cases: 10 feature-length, richly detailed prompts sourced from MoviePrompts (Wu et al., 2025b), and 10 shorter, annotator-designed prompts. For complexity comparison, we focus on the more detailed 10 prompts.

The definitions of the three predefined complexity dimensions are summarized as follows:

- **Semantic Complexity** pertains to the semantic elements within a prompt, including entities and the relationships among them. This dimension necessitates that models accurately interpret the events and interactions in the prompt.

- **Structural Complexity** mainly focuses on the manner in which prompts convey their content, facilitating diverse textual expressions and structured organization. Such complexity challenges models' capacity to process flexible inputs.

- **Control Complexity** concentrates on constraints imposed on the outputs of generative models. Users may, for instance, specify requirements regarding visual style, camera motion, or the presence of specific objects. As such, this dimension is intended to capture these elements in prompts and assess whether models are able to fulfill these requirements.

## B.2. Demonstration of Our Prompt Suite Construction

Due to space constraints, the full visualization of the prompt suite construction process described in 3.2 cannot be included in the main paper. This subsection presents the complete figure, shown in Fig. 7, to facilitate detailed inspection. The figure illustrates the multi-stage workflow of the prompt construction pipeline, including LLM-based generation and subsequent manual verification. The prompt templates for LLM usage in this process could be seen in Appendix D.2.

## B.3. Details about Dynamic Quality

**Implementations of the Sub-dimensions.** Here we provide more detailed descriptions for all the sub-dimensions of dynamic quality in 4.4. Given that these metrics are proposed in VBench (Huang et al., 2024) (**Dynamic Degree** and **Motion Smoothness**) and DEVIL (Liao et al., 2024) (**Patch-level Aperiodicity**, **Global Aperiodicity**, **Information Variance**, and **Temporal Entropy**), we explain the details primarily based on their original settings and our own understanding as follows:

- **Dynamic Degree** evaluates whether a generated video contains observable motion. Its original setting in VBench (Huang et al., 2024) utilizes RAFT (Teed & Deng, 2020) to estimate optical flow between consecutive frames and computes the mean of the top 5% flow magnitudes to classify videos as dynamic or static. Unlike this, we normalize the magnitudes by image resolution and use the average normalized value as a continuous measure for the dynamic degree of the video.

- **Motion Smoothness** evaluates the temporal smoothness of generated video motion. It leverages the motion prior of video frame interpolation models, which assume short-term real-world motion to be approximately linear or quadratic. Given a frame sequence of a generated video $[f_1, f_2, \ldots, f_{2n}]$, all odd-indexed frames are removed to form a low-frame-rate sequence, and an interpolation model is used to reconstruct the missing frames $[\hat{f}_1, \hat{f}_2, \ldots, \hat{f}_{2n-1}]$. The mean absolute error (MAE) between reconstructed and original frames is then computed and normalized as

$$S_{MAE-norm} = \frac{255 - S_{MAE}}{255}. \tag{8}$$

The resulting score lies in $[0, 1]$ with higher values indicating smoother, more physically consistent motion.

- **Patch-level Aperiodicity** calculates inter-segment dynamics at the patch-level. It leverages auto-correlation factor (Box et al., 2015) (ACF), which measures the feature similarity of a time series, to evaluate the scene and temporal pattern dynamics. Given features at position $(h, w)$ across $N$ frames, $\{F_{i,h,w}\}_{i=1}^{N}$ the auto-correlation factor of the features is

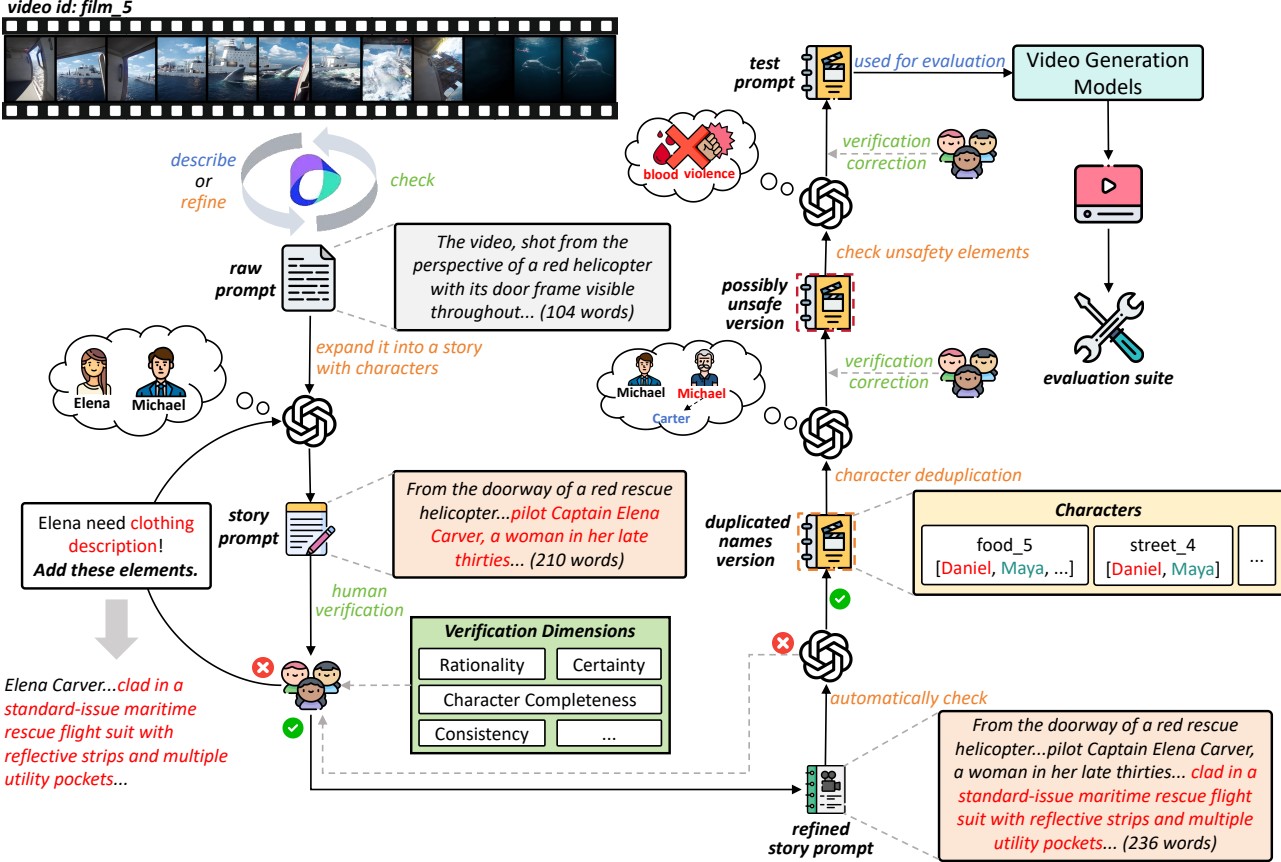

*Figure 7.* **Overview of the example-based workflow for prompt suite construction.** Raw prompts are generated by Seed1.5-VL (Guo et al., 2025), followed by intermediate automatic refinement and inspection using GPT-5 (OpenAI, 2025a).

defined as:

$$\mathbf{ACF}(\{F_{i,h,w}\}_{i=1}^{N}) = \frac{1}{N - K_0} \sum_{k=K_0}^{N} \sum_{i=1}^{k} \frac{1}{k} \mathbf{SIM}(F_{i,h,w}, F_{N-k+i,h,w}), \qquad (9)$$

where $K_0$ is the minimal segment length (empirically set to $\lfloor N/8 \rfloor$) and **SIM** represents the cosine similarity between two feature vectors. $H$ and $W$ refer to the height and width of the feature map, respectively. Then, the patch-level aperiodicity of the video is defined as

$$S_{pa} = 1 - \frac{1}{HW} \sum_{h,w} \mathbf{ACF}(\{F_{i,h,w}\}_{i=1}^{N}). \qquad (10)$$

- **Global Aperiodicity** measures the diversity of patterns between video segments. Assume that each video is divided into segments of length $rN$, where $r$ is a proportion factor empirically set to 0.25. The ViCLIP (Wang et al., 2024) is used to extract the spatial-temporal features for each segment, denoted as $\{F_i^r\}_{i=1}^{\lfloor rN \rfloor}$. Then, the global aperiodicity could be defined as the similarity of these features to assess the variation in spatial-temporal patterns across segments as follows

$$S_{ga} = 1 - \frac{1}{\lfloor rN \rfloor} \sum_{i=1}^{\lfloor rN \rfloor} \sum_{j \neq i} \mathbf{SIM}(F_i^r, F_j^r). \qquad (11)$$

- **Information Variance** could be seen as a semantic diversity score to assess high-level dynamics across the whole video. It's defined as the variance of DINOv2 (Oquab et al.) features $\{F_i\}_{i=1}^{N}$ of each frame as

$$S_{iv} = \frac{1}{N} \sum_{i=1}^{N} ||F_i - \bar{F}||, \qquad (12)$$

where $\bar{F} = \frac{1}{N} \sum_{i=1}^{N} F_i$ denotes the mean feature vector of all frames.

- **Temporal Entropy** measures the temporal information of each video, which is defined as the conditional entropy of the entire video sequence given the first frame

$$S_{te} = \mathbf{H}(f_1, f_2, \cdots, f_N | f_1). \tag{13}$$

The conditional entropy $S_{te}$ is estimated with the assistance of the video encoding toolbox FFmpeg.

**Methodology of Integrating Sub-dimension Scores.** As described in 4.4, we employ a pre-fitted linear model to aggregate the sub-dimension scores of dynamic quality. This model is obtained by fitting on a randomly sampled set of 100 videos, which are annotated by the same three annotators described in Appendix B.6. Notably, these samples are disjoint from those used in the human-alignment analysis in 5.2. We use this model to compute the overall dynamic quality score in all related evaluations.

### B.4. Details of HERD Evaluation

As mentioned in 4.5, we employ a dual-agent framework to evaluate the HERD metric across its six sub-dimensions. The prompts used in this framework could be seen in Appendix D.4. We also give a detailed introduction about all dimensions included in HERD metrics as follows:

- **Emotional Response** assesses the emotional impact of the video—whether it evokes curiosity, tension, inspiration, or confusion—and examines how effectively it engages viewers' feelings and maintains their emotional attention throughout. (e.g., *The video is expected to evoke wonder, serenity, and a sense of cosmic discovery. Viewers should feel...*)

- **Narrative Flow** examines the clarity and coherence of the storyline, including scene transitions and pacing, focusing on whether the narrative unfolds smoothly, feels rushed, or allows moments for reflection. (e.g., *The storytelling should progress smoothly from exploration to reflection, following...*)

- **Character Development** evaluates the depth, authenticity, and consistency of characters, as well as the evolution of their relationships, emphasizing how these elements contribute to audience engagement and narrative believability. (e.g., *The woman is expected to embody curiosity...*)

- **Visual Style** analyzes the use of cinematography, color palette, lighting, and framing in establishing mood, atmosphere, and tone, considering how visual choices enhance story immersion and emotional resonance. (e.g., The video should feature a lush cosmic aesthetic...)

- **Themes Expression** reflects on the underlying ideas, messages, or social commentary, assessing whether they are clearly expressed, thought-provoking, and meaningfully integrated with the video's overall narrative and intent. (e.g., Core ideas of exploration, observation, and...)

- **Overall Impression** captures the lasting effect of the video, considering its overall impact, memorability, and appeal, and reflecting on its entertainment, educational, or emotional value for a broad range of audiences. (e.g., The video should leave a lasting sense of peace and...)

Additionally, these HERD dimensions are also grounded in established theories from **Cognitive Film Theory** and **Narratology**. Specifically, *Narrative Flow* and *Character Development* draw upon **Classical Film Narratology** (Bordwell, 2013) and **Character Engagement Theory** (Smith, 2022) to evaluate the structural and psychological coherence of the generated narrative. *Visual Style* relates to the cinematic concept of *mise-en-scène* (Bordwell et al., 2008), focusing on how visual attributes such as lighting and framing jointly establish mood and atmosphere. *Emotional Response* is motivated by **Aesthetic Appraisal Models** (Leder et al., 2004), measuring the emotional and aesthetic impact of the generated visuals. Finally, *Themes* and *Overall Impression* are informed by **Cinematic Semiotics** (Metz, 1991), assessing the higher-level semantic interpretation and synthesis of cinematic elements.

### B.5. Introduction & Implementations about Baseline Methods

**Scene-level Prompts Division.** As shown in Table 2, some of the selected long video generation (LVG) methods support only multi-prompt inputs, requiring a separate prompt for each scene during generation. To fairly evaluate these methods within our framework, we adopt a scene-level prompt division strategy based on a generate-then-verify paradigm, as illustrated in Fig. 8. Specifically, we first employ GPT-5 (OpenAI, 2025a) to extract scene-level summaries from the original

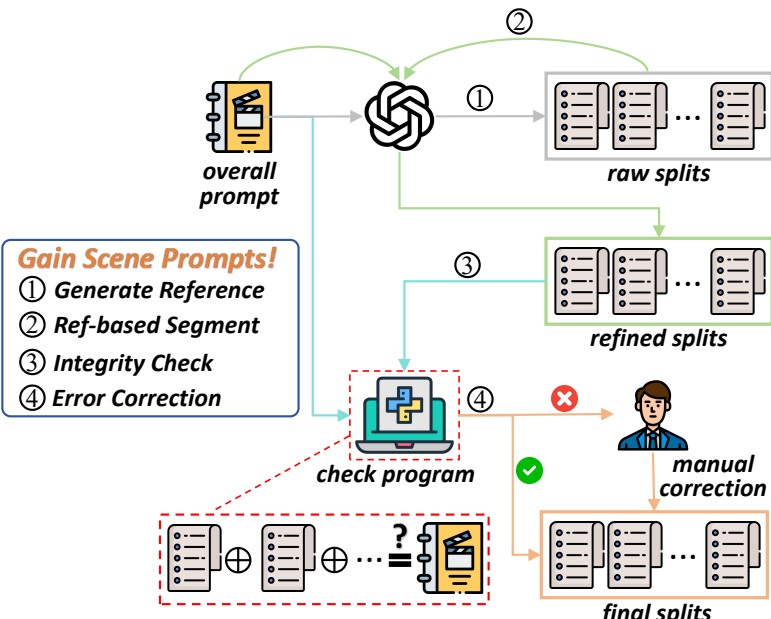

*Figure 8.* **Process of scene-level prompts division.** We complete this process in four steps with the help of GPT-5 (OpenAI, 2025a), program-based automatic verification and manual correction.

prompt (e.g., *"A man stands on the shore watching the giant planet above the water."*). These summaries are then refined to ensure strict faithfulness to the original prompt. To ensure a fair comparison between multi-prompt-based methods and those that accept a single, direct input prompt, the overall semantic content of the inputs must remain consistent. To this end, we design an automated verification procedure that checks whether the direct concatenation of all scene-level prompts exactly reconstructs the original prompt. Invalid cases are further manually corrected to guarantee the integrity and consistency of the final evaluation results.

**Baseline Introductions.** For a comprehensive evaluation of current long video generation techniques, we select 13 representative methods based on their availability, popularity in the community, and diversity in modeling strategies. The selected methods are listed as follows:

- **FreeNoise** (Qiu et al., 2024) proposes a tuning-free paradigm for longer video generation with pretrained diffusion models by rescheduling initial noise to maintain long-range temporal coherence, plus a motion-injection trick to support multi-prompt conditioning, achieving superior quality.

- **MEVG** (Oh et al., 2024) is a training-free pipeline that turns a pre-trained T2V diffusion model into a multi-prompt storyteller. It uses an LLM prompt generator to split a long story into single-event captions and injects dynamic noise and last-frame inversion to initialize each new clip from the previous last frame, then applies structure-guided sampling to keep frames within a clip coherent.

- **FreeLong** (Lu et al., 2024) proposes a training-free SpectralBlend Temporal Attention mechanism: it fuses the low-frequency parts of global features (for overall coherence) with the high-frequency parts of local features (for fine detail) via 3-D FFT, enabling a 16-frame diffusion model to generate 128-frame videos with better consistency and fidelity.

- **FIFO-Diffusion** (Kim et al., 2024) enables a pretrained short-clip diffusion model to generate endless videos without retraining by performing diagonal denoising in a small FIFO frame queue, where noise increases toward the tail while the clean head is popped and new noise is pushed. To bridge the gap with uniform-noise training and reduce memory usage, it further introduces latent partitioning and lookahead denoising, achieving high-quality, temporally coherent long video generation.

- **DiTCtrl** (Cai et al., 2025) proposes a tuning-free approach for long video generation from multiple text prompts based on the MM-DiT architecture. By analyzing and leveraging its attention mechanism, it achieves smooth transitions and consistent motion via a novel KV-sharing strategy and latent blending.

- **StoryAdatper** (Mao et al., 2024) proposes a training-free iterative framework for long-story visualization based on

pretrained Stable Diffusion models (Rombach et al., 2022). By leveraging a Global Reference Cross-Attention (GRCA) mechanism that incorporates previously generated images as global references, it achieves strong semantic consistency and fine-grained interactions, even for long stories of up to 100 frames. However, since StoryAdapter produces only scene-level images rather than continuous videos, we employ a powerful image-to-video (I2V) model, Wan2.2 (Wan et al., 2025), to generate a short clip for each scene and concatenate them to obtain the final long video.

- **CausVid** (Yin et al., 2025) is a fast autoregressive video diffusion model distilled from a slow bidirectional teacher using asymmetric distribution matching distillation (DMD), reducing generation latency from 219 s to 1.3 s and enabling streaming 9.4 FPS video on one GPU while maintaining state-of-the-art quality.

- **SkyReels-V2** (Chen et al., 2025a) synergizes an MLLM-based captioner, multi-stage pre-training, motion-specific reinforcement learning, and a diffusion-forcing framework to generate infinite-length, cinematic-quality videos while achieving state-of-the-art prompt adherence and motion fidelity. We use its 540P version in our practice.

- **Vlogger** (Zhuang et al., 2024) proposes an LLM-directed pipeline that decomposes a long vlog into four stages—Script, Actor, ShowMaker, Voicer—and introduces a new diffusion model (ShowMaker) that conditions on both text and actor images to generate coherent, variable-length scenes. It can produce 5-minute vlogs from open-world text without extra long-video training, setting a new zero-shot baseline for long video generation.

- **VGoT** (Zheng et al., 2024) is a training-free modular framework for multi-shot video generation. It decomposes the process into four collaborative modules: script generation, keyframe creation, shot-level video synthesis, and cross-shot smoothing. It ensures narrative coherence and visual consistency across shots using structured cinematic prompts and identity-preserving embeddings.

- **SANA-Video** (Chen et al., 2025b) introduces an efficient video generation framework based on Linear Diffusion Transformer (DiT) (Xie et al., 2025b) with linear attention complexity. By introducing Block Linear Attention with a constant-memory KV cache derived from cumulative properties, it achieves high-resolution (720p) and minute-length video generation with dramatically reduced computational costs—training in just 12 days on 64 H100 GPUs (1% of MovieGen's cost (Polyak et al., 2024)) while being 16× faster in inference than comparable models.

- **LongLive** (Yang et al., 2025a) presents a frame-level autoregressive framework for real-time interactive long video generation. By introducing a KV-recache mechanism for smooth prompt switching, streaming long tuning for train-test alignment, and short window attention with frame-level attention sink for efficiency, it achieves 20.7 FPS on a single H100 GPU and supports up to 240-second videos with only 32 GPU-days of fine-tuning.

- **LongSANA** (Chen et al., 2025b) is the specialized long-video variant of the SANA-Video and is designed to create minute-long, high-resolution videos in real time, which could be seen as an integration of SANA-Video and LongLive.

- **LongCat-Video** (Team et al., 2025) is a 13.6B-parameter foundational DiT-based model designed for efficient and high-quality long video generation. It employs a unified architecture that seamlessly supports text-to-video, image-to-video, and video-continuation tasks. To ensure temporal coherence and efficient minutes-long synthesis at 720p resolution, it leverages a spatiotemporal coarse-to-fine strategy, block sparse attention, and multi-reward RLHF.

- **Sora 2** (OpenAI, 2025b) is OpenAI's flagship audio-video generation model designed to serve as an advanced simulator of the physical world. It achieves a qualitative leap in physical accuracy, visual realism, and multi-shot steerability, seamlessly synthesizing high-fidelity videos with synchronized dialogue and complex sound effects. Furthermore, it ensures world-state consistency and enables precise, zero-shot identity preservation through its innovative "Character" feature across diverse stylistic domains.

- **Seedance 1.5-Pro** (Seedance et al., 2025) advances unified audio-visual synthesis through a dual-branch DiT that intrinsically couples video and audio generation. By leveraging a specialized cross-modal joint module and high-quality SFT alongside RLHF, it achieves state-of-the-art audio-visual synchronization. Notably, it supports highly efficient inference with a 10× speedup while excelling in precise multilingual and dialect lip-syncing.

- **Kling 3.0** (Kuaishou, 2025) is an advanced unified multimodal foundation model that natively integrates multiple generative tasks, including text-to-video and image-to-video synthesis. It achieves a qualitative leap in AI storytelling by introducing intelligent multi-shot generation for up to 15-second narratives, alongside a native cross-modal audio engine for highly synchronized, multilingual dialogues. Furthermore, it employs a robust multimodal reference and decoupling control mechanism to guarantee exceptional spatiotemporal consistency and precise character identity preservation across complex scenes.

**Baseline Generation Settings.** To ensure a fair comparison, we standardize key parameters that may influence video generation quality across all baselines. Specifically, we fix the random seed to 0, the frame rate to 16 fps, and the output resolution to 480p. However, certain methods—such as SkyReels-V2 (Chen et al., 2025a), Vlogger (Zhuang et al., 2024), and VGoT (Zheng et al., 2024)—do not perform reliably at 480p. Moreover, for models with audio-visual generation capabilities like Sora2, we disable their audio generation and only evaluate the generated videos during our assessment.

## B.6. Human Annotation

As described in 5.2, we assess the alignment between our evaluation results and human preferences. This section details the human annotation procedure. We sample 5% of all generated test videos for annotation and recruit three experienced human annotators. Annotators are instructed to assign scores on a 1-5 scale for the five dimensions defined in 4, or to answer the binary questions introduced in 4.2, depending on the evaluation setting. Moreover, annotators are not required to assign scores to each individual sub-dimension of LoCoT2V-Eval. Instead, they provide an overall score for dynamic quality and evaluate character and background consistency within the temporal quality dimension. These aspects are selected for human annotation as they are more intuitive and reliable for annotators than semantic consistency or warping error. For each sample, we compute the Mean Opinion Score (MOS) by averaging the scores across annotators. Finally, we measure the correlation between the human annotation results and the corresponding scores produced by our evaluation framework, LoCoT2V-Eval. An example of the annotation interface is shown in Fig. 9.

## B.7. Implementations of Other HERD Evaluation Methods

In 5.2 we mention two other feasible HERD evaluation methods; here we provide details about them.

**QA-based HERD Evaluation.** Given that the HERD expectations take the form of requirement-like textual descriptions (see Appendix B.4), a natural alternative evaluation strategy is to convert them into corresponding binary questions (e.g., *"Does this video evoke a sense of humor?"*) and then employ Qwen3-VL-8B (Bai et al., 2025a) to determine whether a generated video satisfies each expectation. The final score can be computed as the proportion of positive ("Yes") responses. To compare against this baseline, we first use GPT-5 (OpenAI, 2025a) to generate binary questions from the HERD expectations (the prompt used for this step is provided in Appendix D.4). We then apply the same MLLM as used in our dual-agent evaluation framework described in 4.5 to answer these questions. The score for this method is defined as the proportion of "Yes" responses. We further manually verify the polarity of each question to ensure that a "Yes" answer consistently contributes positively to the final score.

**Direct HERD Scoring.** Another alternative approach is to directly employ an MLLM to evaluate generated videos according to the definitions of each HERD dimension. Specifically, we prompt the same MLLM used in 4.5 to assign a score ranging from 1 to 5 for each video along every HERD dimension (See our used prompt in Appendix D.4). These scores are subsequently normalized to the range $[0, 1]$ by dividing by the maximum possible score of 5.

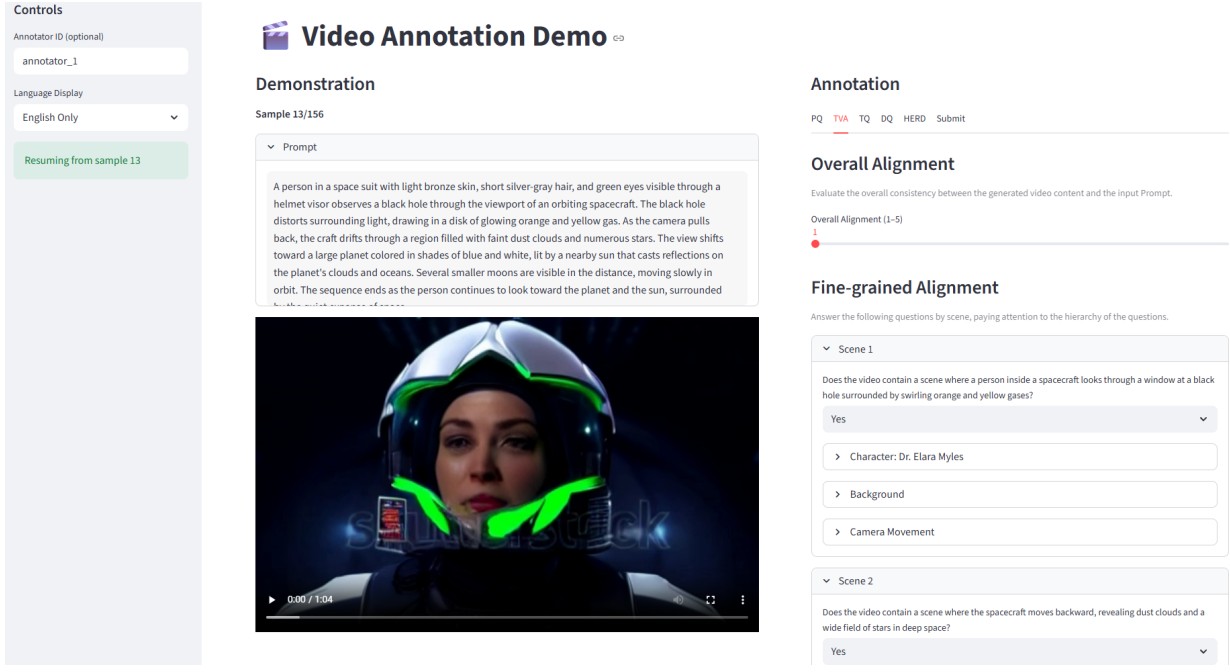

*Figure 9.* **Demonstration of our designed demo for human annotation.** Annotators are supposed to assign scores in five dimensions based on their preference, and some of the sub-dimensions are also included in this process.

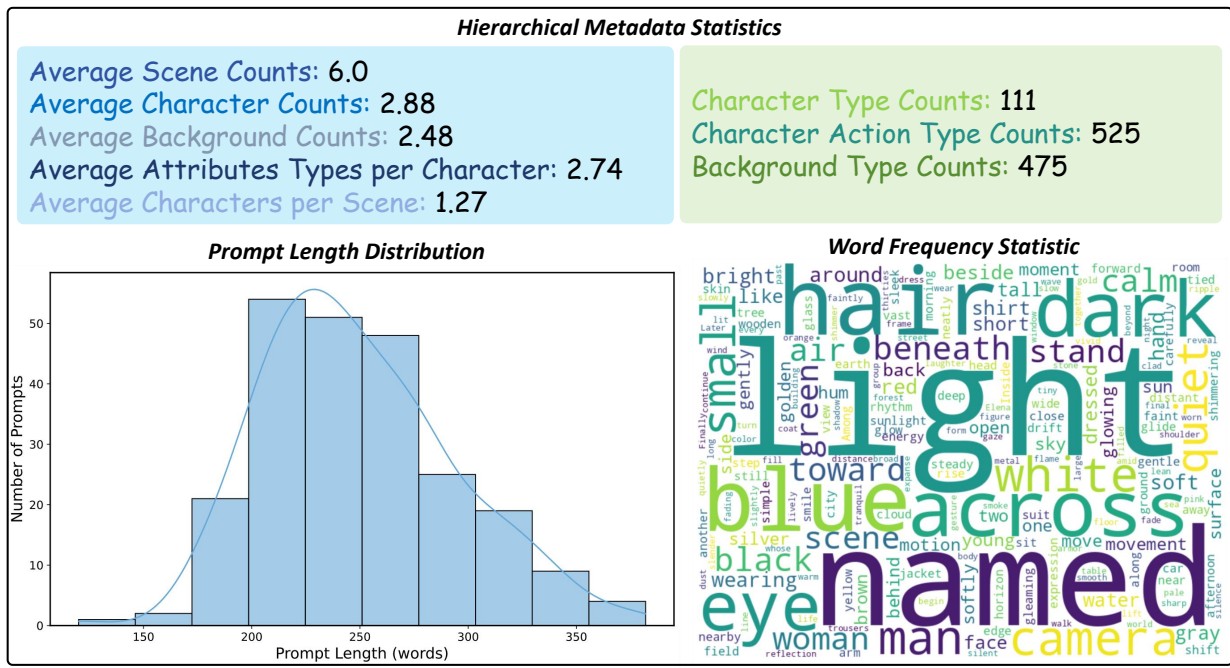

*Figure 10.* **Other Prompt Suite Statistics.** The two graphs demonstrate some other statistics of our prompt suite. *left:* the prompt length distribution of our prompt suite measured by the number of words. *right:* the word cloud to visualize word distribution of our prompts.

*Table 5.* **Performance results of all baseline methods on five evaluation dimensions.** Samples are grouped into three categories according to prompt content themes, and the performance of each group is reported for every method. Note that all values are expressed as percentages to improve readability and conserve space.

| Category | Method | Perceptual Quality | Text-Video Alignment | Temporal Quality | HERD | Dynamic Quality | Avg. |
|---|---|---|---|---|---|---|---|
| Daily Life | FreeNoise (Qiu et al., 2024) | 73.95 | 14.04 | 71.54 | 54.21 | 52.14 | 53.18 |
| | MEVG (Oh et al., 2024) | 42.23 | 16.00 | 62.44 | 45.76 | 33.42 | 39.97 |
| | FreeLong (Lu et al., 2024) | 70.26 | 25.20 | 71.54 | 59.83 | 45.67 | 54.50 |
| | DiTCtrl (Cai et al., 2025) | 57.08 | 44.69 | 73.99 | 60.51 | 47.37 | 56.73 |
| | StoryAdapter (Mao et al., 2024) | 80.70 | 45.53 | 77.90 | 72.05 | 50.34 | 65.30 |
| | CausVid (Yin et al., 2025) | 68.08 | 37.18 | _83.43_ | 69.23 | 58.12 | 63.21 |
| | FIFO-Diffusion (Kim et al., 2024) | 71.49 | 12.65 | 69.85 | 69.23 | 53.11 | 55.27 |
| | SkyReels-V2 (Chen et al., 2025a) | 73.11 | 49.85 | 76.68 | 88.15 | 61.40 | 69.84 |
| | Vlogger (Zhuang et al., 2024) | 64.27 | 32.25 | 67.97 | 63.03 | 39.19 | 53.34 |
| | VGoT (Zheng et al., 2024) | _83.44_ | 39.32 | 78.22 | 69.49 | 60.43 | 66.18 |
| | SANA-Video (Chen et al., 2025b) | 75.80 | 36.58 | 75.41 | 73.57 | 60.11 | 64.29 |
| | LongLive (Yang et al., 2025a) | 81.39 | 48.39 | **87.63** | 88.99 | _62.04_ | _73.69_ |
| | LongSANA (Chen et al., 2025b) | **84.37** | 31.57 | 82.23 | 83.16 | 60.24 | 68.31 |
| | LongCat-Video (Team et al., 2025) | 79.58 | 58.83 | 81.06 | **93.30** | 59.00 | **74.35** |
| | Sora2 (OpenAI, 2025b) | 67.54 | _61.07_ | 83.17 | 89.61 | **65.39** | 73.36 |
| | Seedance 1.5-Pro (Seedance et al., 2025) | 71.44 | 56.26 | 80.75 | _92.02_ | 57.29 | 71.55 |
| | Kling 3.0 (Kuaishou, 2025) | 71.68 | **64.96** | 79.65 | 91.12 | 56.61 | 72.80 |
| Nature | FreeNoise (Qiu et al., 2024) | 74.07 | 16.75 | 68.37 | 54.91 | 47.93 | 52.41 |
| | MEVG (Oh et al., 2024) | 40.07 | 21.01 | 62.38 | 47.55 | 33.25 | 40.85 |
| | FreeLong (Lu et al., 2024) | 65.62 | 28.95 | 71.11 | 60.00 | 46.08 | 54.35 |
| | DiTCtrl (Cai et al., 2025) | 60.29 | 46.86 | 71.31 | 65.97 | 50.99 | 59.08 |
| | StoryAdapter (Mao et al., 2024) | 82.30 | 46.61 | **80.38** | 69.81 | 51.27 | 66.07 |
| | CausVid (Yin et al., 2025) | 71.38 | 38.31 | 75.98 | 73.01 | 57.63 | 63.26 |
| | FIFO-Diffusion (Kim et al., 2024) | 71.58 | 13.31 | 68.62 | 69.12 | 51.78 | 54.88 |
| | SkyReels-V2 (Chen et al., 2025a) | 71.07 | 36.23 | 72.94 | 78.43 | 59.56 | 63.65 |
| | Vlogger (Zhuang et al., 2024) | 64.11 | 38.53 | 68.39 | 61.81 | 42.48 | 55.06 |
| | VGoT (Zheng et al., 2024) | _83.17_ | 42.21 | 75.71 | 67.92 | 59.54 | 65.71 |
| | SANA-Video (Chen et al., 2025b) | 72.11 | 35.57 | 70.76 | 69.21 | 58.64 | 61.26 |
| | LongLive (Yang et al., 2025a) | 81.28 | 43.93 | _79.95_ | 80.37 | _60.59_ | 69.22 |
| | LongSANA (Chen et al., 2025b) | **84.04** | 31.79 | 76.27 | 77.64 | 58.91 | 65.73 |
| | LongCat-Video (Team et al., 2025) | 79.58 | 54.59 | 75.82 | 83.10 | 59.47 | 70.51 |
| | Sora2 (OpenAI, 2025b) | 68.80 | **65.77** | 77.94 | _85.56_ | **64.27** | **72.47** |
| | Seedance 1.5-Pro (Seedance et al., 2025) | 71.33 | 58.17 | 77.00 | 84.88 | 57.37 | 69.75 |
| | Kling 3.0 (Kuaishou, 2025) | 71.87 | _63.18_ | 76.04 | **87.45** | 56.51 | _71.01_ |
| Virtual | FreeNoise (Qiu et al., 2024) | 73.61 | 11.73 | 68.87 | 51.32 | 51.07 | 51.32 |
| | MEVG (Oh et al., 2024) | 39.17 | 13.78 | 61.21 | 40.63 | 34.91 | 37.94 |
| | FreeLong (Lu et al., 2024) | 62.66 | 22.71 | 66.99 | 52.96 | 46.88 | 50.44 |
| | DiTCtrl (Cai et al., 2025) | 51.45 | 50.40 | 71.53 | 55.19 | 50.76 | 55.87 |
| | StoryAdapter (Mao et al., 2024) | 74.87 | 40.27 | 74.28 | 62.38 | 49.27 | 60.21 |
| | CausVid (Yin et al., 2025) | 68.59 | 30.93 | 80.07 | 62.33 | 58.50 | 60.08 |
| | FIFO-Diffusion (Kim et al., 2024) | 69.35 | 9.15 | 69.70 | 60.58 | 56.84 | 53.12 |
| | SkyReels-V2 (Chen et al., 2025a) | 65.47 | 49.36 | 73.08 | 67.35 | 59.20 | 62.89 |
| | Vlogger (Zhuang et al., 2024) | 59.57 | 29.27 | 67.40 | 54.02 | 43.60 | 50.77 |
| | VGoT (Zheng et al., 2024) | _81.11_ | 37.32 | 74.95 | 65.34 | 59.16 | 63.58 |
| | SANA-Video (Chen et al., 2025b) | 73.29 | 28.04 | 70.01 | 67.83 | 58.70 | 59.57 |
| | LongLive (Yang et al., 2025a) | 78.25 | 43.96 | **81.67** | 70.26 | _61.65_ | 67.16 |
| | LongSANA (Chen et al., 2025b) | **83.80** | 24.69 | 76.33 | 72.91 | 59.85 | 63.52 |
| | LongCat-Video (Team et al., 2025) | 72.78 | _61.72_ | 77.36 | 73.39 | 59.53 | 68.96 |
| | Sora2 (OpenAI, 2025b) | 61.75 | 57.98 | _81.02_ | _81.60_ | **64.32** | _69.33_ |
| | Seedance 1.5-Pro (Seedance et al., 2025) | 68.86 | 58.23 | 76.16 | 79.31 | 56.91 | 67.89 |
| | Kling 3.0 (Kuaishou, 2025) | 66.21 | **67.17** | 79.72 | **81.80** | 55.06 | **69.99** |

# C. Complementary Results

## C.1. More Statistics for the Prompts in LoCoT2V-Bench

In addition to the overview presented in Fig. 1, we report further statistics of the LoCoT2V-Bench prompt suite in Fig. 10. These statistics provide complementary insights into the properties of our prompts. First, while character-related descriptions constitute a substantial portion of the prompt content, a wide range of action types and background categories is also covered, ensuring prompt diversity. Second, each sample contains six scenes on average, often involving multiple characters or backgrounds, which reflects the structural complexity of the prompts. Third, the prompt lengths predominantly fall within the range of $[150, 350]$, substantially exceeding the average prompt length of existing benchmarks reported in Appendix B.1. Collectively, these statistics further characterize the scale, diversity, and complexity of our prompt suite.

## C.2. Category-level Prompt Statistics

Given that our prompts are organized into 18 fine-grained themes under three high-level categories, as illustrated in Fig. 1, we further partition the test samples into three groups according to the prompt categories and report category-level results to examine the relationship between prompt categories and method performance.

We first present the average prompt length and complexity scores for each category in Fig. 11. As shown, the prompt complexity exhibits minor variation across these categories. In contrast, the prompt content category has a stronger influence on prompt length: the *"Virtual"* category contains the longest prompts, whereas the *"Nature category"* has the shortest ones.

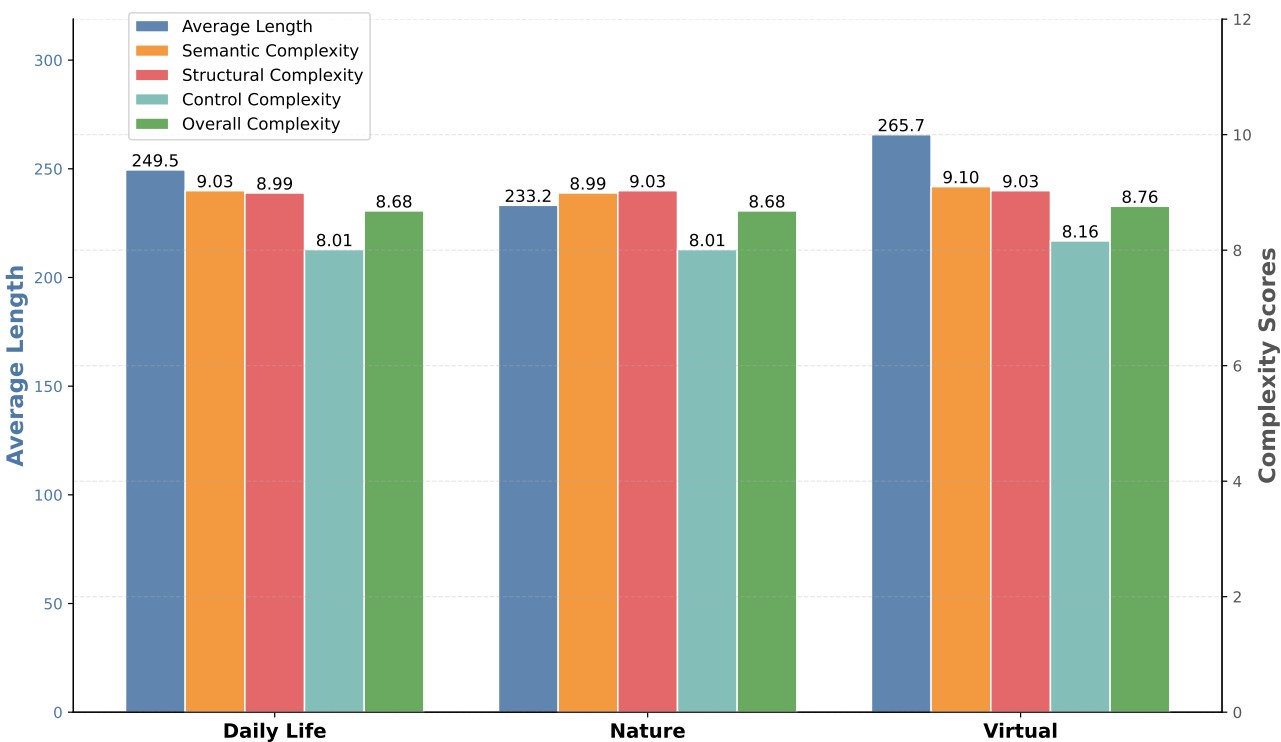

*Figure 11.* **Category-level statistics about the average length and complexity scores of the prompts.**

## C.3. Category-level Results of Baseline Methods

Here, we present the category-level results of all baseline methods across the five defined evaluation dimensions. For each dimension, the reported score is computed as the average over its corresponding sub-dimensions. The results are summarized in Table 5. We observe that the relative performance trends among methods at the category level remain largely consistent with the overall results reported in Table 2. This consistency further supports the robustness of our evaluation framework,

despite the imbalanced number of samples across categories as shown in Fig. 1.

## C.4. Influence of the Prompt Complexity

As described in B.1, we define several metrics to characterize the complexity of our prompts (e.g., semantic complexity). We further investigate how these complexity metrics relate to model performance across the five evaluation dimensions via correlation analysis, as shown in Fig. 12.

From Fig. 12 we can observe that all complexity metrics exhibit consistently negative but weak correlations with model performance on the five dimensions. We attribute the weak correlations to the limited variance of these metrics (including prompt length), as reported in Table 6. Nevertheless, the negative correlation trend aligns with intuition, providing additional support for the validity of our complexity metric definitions.

*Table 6.* **Variance of each complexity metric.**

| Metric | Variance (%) |
| --- | --- |
| Semantic Complexity | 2.87 |
| Structural Complexity | 2.99 |
| Control Complexity | 4.40 |
| Overall Complexity | 2.84 |
| Prompt Length | **17.76** |

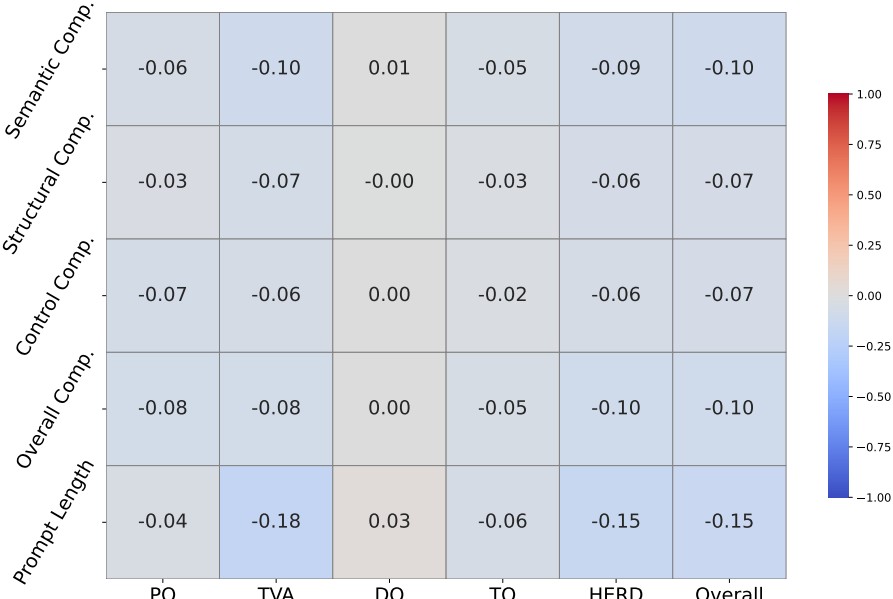

*Figure 12.* **Correlation results between different complexity metrics and the five major evaluation dimensions.** Scores are computed on all generated samples by baseline models. Note that *Comp.* refers to "*Complexity*" in this figure.

## C.5. Results for Sub-dimensions of Dynamic Quality and HERD

Here we provide scores of all sub-dimensions of **Dynamic Quality** (Table 7) and **HERD** (Table 8), respectively. These results could be seen as a supplement to those scores in Table 2.

*Table 7.* **Results on all sub-dimensions of dynamic quality.**

| Method | Frame-level | | Segment-level | | Video-level | | Overall |
|---|---|---|---|---|---|---|---|
| | Motion Smoothness | Dynamic Degree | Patch-level Aperiodicity | Global Aperiodicity | Information Variance | Temporal Entropy | |
| FreeNoise (Qiu et al., 2024) | 97.64 | 29.11 | 75.10 | 46.57 | 49.79 | 64.72 | 50.55 |
| MEVG (Oh et al., 2024) | 96.14 | 39.96 | 85.57 | 63.06 | 68.23 | 22.39 | 33.78 |
| FreeLong (Lu et al., 2024) | 97.33 | 32.64 | 84.38 | 62.99 | 67.21 | 59.45 | 46.10 |
| DiTCtrl (Cai et al., 2025) | 97.99 | 63.94 | 88.40 | 63.97 | 67.52 | 44.42 | 49.37 |
| StoryAdapter (Mao et al., 2024) | 97.66 | 64.49 | 87.52 | 59.67 | 63.06 | 57.97 | 50.36 |
| CausVid (Yin et al., 2025) | 98.21 | 65.82 | 76.37 | 41.84 | 44.21 | 72.28 | 58.06 |
| FIFO-Diffusion (Kim et al., 2024) | 96.62 | 77.33 | 84.39 | 39.99 | 47.07 | 86.43 | 53.72 |
| SkyReels-V2 (Chen et al., 2025a) | 98.54 | 65.85 | 81.33 | 46.29 | 51.73 | 72.30 | 60.24 |
| Vlogger (Zhuang et al., 2024) | 95.95 | 44.50 | **91.37** | **69.32** | **77.09** | 74.71 | 41.37 |
| VGoT (Zheng et al., 2024) | **99.11** | 37.21 | 87.91 | 58.17 | 62.54 | 70.52 | 59.83 |
| SANA-Video (Chen et al., 2025b) | 98.09 | 60.23 | 79.03 | 42.86 | 50.70 | 79.22 | 59.30 |
| LongLive (Yang et al., 2025a) | 98.69 | 52.91 | 75.66 | 36.71 | 44.04 | 69.44 | 61.52 |
| LongSANA (Chen et al., 2025b) | 99.01 | 38.09 | 71.25 | 32.97 | 36.76 | 54.80 | 59.73 |
| LongCat-Video (Team et al., 2025) | 98.01 | 85.08 | 86.73 | 49.45 | 56.48 | 77.38 | 59.29 |
| Sora2 (OpenAI, 2025b) | 99.05 | 49.21 | 77.66 | 47.19 | 51.30 | **92.15** | **64.78** |
| Seedance 1.5-Pro (Seedance et al., 2025) | 98.88 | **90.21** | 85.00 | 50.51 | 53.26 | 39.00 | 57.21 |
| Kling 3.0 (Kuaishou, 2025) | 98.87 | 81.31 | 83.13 | 52.41 | 54.97 | 39.53 | 56.16 |

*Table 8.* **Results on all sub-dimensions of HERD.**

| Method | Emotional Response | Narrative Flow | Character Development | Visual Style | Themes Expression | Overall Impression | Avg. |
|---|---|---|---|---|---|---|---|
| FreeNoise (Qiu et al., 2024) | 55.38 | 49.49 | 37.78 | 71.28 | 55.13 | 52.82 | 53.65 |
| MEVG (Oh et al., 2024) | 48.29 | 36.75 | 27.78 | 65.13 | 47.61 | 44.02 | 44.93 |
| FreeLong (Lu et al., 2024) | 61.79 | 45.30 | 44.02 | 77.26 | 61.45 | 58.38 | 58.03 |
| DiTCtrl (Cai et al., 2025) | 64.53 | 53.68 | 49.74 | 71.79 | 65.04 | 59.74 | 60.75 |
| StoryAdapter (Mao et al., 2024) | 71.88 | 55.64 | 60.09 | 84.19 | 71.71 | 69.06 | 68.76 |
| CausVid (Yin et al., 2025) | 68.89 | 68.80 | 54.96 | 79.57 | 70.26 | 68.72 | 68.53 |
| FIFO-Diffusion (Kim et al., 2024) | 68.12 | 63.93 | 51.20 | 83.25 | 67.86 | 66.84 | 66.87 |
| SkyReels-V2 (Chen et al., 2025a) | 78.72 | 78.80 | 70.09 | 88.38 | 81.11 | 80.26 | 79.56 |
| Vlogger (Zhuang et al., 2024) | 62.05 | 48.12 | 51.28 | 76.07 | 62.74 | 61.11 | 60.23 |
| VGoT (Zheng et al., 2024) | 69.32 | 53.50 | 61.97 | 84.44 | 69.83 | 68.29 | 67.89 |
| SANA-Video (Chen et al., 2025b) | 70.26 | 69.32 | 58.72 | 82.82 | 71.88 | 71.11 | 70.68 |
| LongLive (Yang et al., 2025a) | 79.49 | 80.77 | 72.56 | 90.51 | 82.82 | 81.62 | 81.30 |
| LongSANA (Chen et al., 2025b) | 77.61 | 76.75 | 66.50 | 91.79 | 80.43 | 79.15 | 78.70 |
| LongCat-Video (Team et al., 2025) | 83.85 | 82.48 | 79.49 | 91.45 | 86.24 | 85.30 | 84.80 |
| Sora2 (OpenAI, 2025b) | 84.79 | 84.23 | 79.44 | **94.74** | 88.73 | 86.57 | 86.42 |
| Seedance 1.5-Pro (Seedance et al., 2025) | 85.15 | 84.98 | 80.34 | 93.22 | 87.90 | 86.87 | 86.41 |
| Kling 3.0 (Kuaishou, 2025) | **86.44** | **86.01** | **81.63** | 93.65 | **89.27** | **87.81** | **87.47** |

# D. Prompts Used in Our Practice

### D.1. Complexity Scoring Prompt

**Complexity Scoring Prompt**

```
You are an expert evaluator of prompts used for image or video generation.
Your task is to analyze the complexity of a given prompt in detail.
Return the output strictly as a JSON object in the following nested dictionary
structure:
{
    "semantic_complexity": {
        "score": <integer 1-10>,
```

```
        "explanation": "<short explanation>"
    },
    "structural_complexity": {
        "score": <integer 1-10>,
        "explanation": "<short explanation>"
    },
    "control_complexity": {
        "score": <integer 1-10>,
        "explanation": "<short explanation>"
    }
}

### Evaluation criteria ###

1. Semantic complexity:
    - Number of entities (subjects, objects, characters).
    - Number of attributes or modifiers.
    - Abstract or metaphorical concepts.
    - Relationships or interactions between entities.

2. Structural complexity:
    - Prompt length and density.
    - Nested or hierarchical descriptions.
    - Logical relations (conditions, causality, comparisons).
    - Scene richness (multiple settings or sub-elements).

3. Control complexity:
    - Artistic or stylistic constraints (anime, cyberpunk, Van Gogh, etc.).
    - Technical constraints (camera angle, lens type, lighting).
    - Temporal dynamics (video actions, transitions).
    - Consistency requirements (identity or object continuity).
    - Explicit numeric or technical parameters.

### Few-shot examples ###

**Example 1 (simple prompt):**
[Example 1]

**Example 2 (moderately complex prompt):**
[Example 2]

**Example 3 (highly complex prompt):**
[Example 3]

### Now evaluate the following prompt:
{prompt_text}
```

## D.2. Prompt Suite Construction Related Prompts

**Self-Refine: Video Description Generation Prompt**

```
## System ##
You are a highly capable visual understanding assistant, skilled in analyzing and
summarizing video content with precision and clarity.

## Task ##
Your goal is to produce a coherent and clear paragraph that accurately summarizes the
content of a given video.

Please follow these steps internally (do not output intermediate results):
```

1. **Event Detection**: Identify all major events in the video and arrange them in chronological order.
2. **Visual Element Analysis**: For each event, identify and deeply analyze the key visual components by specifying their attributes and visual characteristics:
   - Subjects: Identify each subject (e.g., person, animal, object) and describe their appearance (e.g., clothing, facial expression, posture, size, color, design).
   - Environments: Describe the setting in detail, including lighting conditions, spatial layout, textures, atmosphere, and any notable background elements.
   - Actions: Detail the actions with clarity-specify how movements are performed (e.g., slow vs. rapid, smooth vs. abrupt), gestures, and interaction between subjects or with objects.
   - Camera Dynamics: Describe how the camera behaves visually-note the type, speed, and purpose of camera movements (e.g., a slow pan to build suspense, a sudden zoom to highlight surprise), including angle perspectives and focal changes.
3. **Event Description**: Describe each event accurately, incorporating the visual elements identified.
4. **Summary Composition**: Integrate all event descriptions into a single, well-structured paragraph that captures the full sequence and essence of the video.

## Output Format ##
Only output the final summary paragraph. Do not include any intermediate steps, bullet points, or reasoning process.

---

## Self-Refine: Description Check Prompt

## System
You are a precise and critical video-reviewing assistant. Your task is to evaluate a paragraph that describes a video, based on the actual video content.

## Task
Given a video and a textual description, assess whether the description accurately and thoroughly reflects the video content.

### If the description is accurate and complete:
- Only output the sentence: The description is good.

### If there are issues:
- Please only output a **numbered list** of specific and constructive revision suggestions. No need to give the revised result or other intermediate results.
- Focus your comments on the following aspects:
  - Does the description match the actual **events** shown in the video?
  - Are the **subjects**, **settings**, **actions**, and **camera motion** well-described and aligned with what appears?
  - Are there any missing or misrepresented visual details?

Be specific, concise, and do not suggest stylistic changes unrelated to factual alignment or visual accuracy.

Here is the description:
{description_text}

---

## Self-Refine: Video Description Refine Prompt

## System
You are a skilled text editor focused on factual and visual alignment.

## Task
You will receive:

```
1. A paragraph that describes a video.
2. A set of revision suggestions.

Your job is to **revise the description** to address all points in the suggestions,
while keeping the paragraph coherent and faithful to the original style.

### Requirements:
- Implement all factual and visual corrections.
- Ensure the updated paragraph fully and accurately reflects the events in the video,
including:
  - Who or what is involved (**subjects**)
  - Where the events take place (**setting**)
  - What happens (**actions**)
  - How the camera moves or frames the scenes (**camera motion**)
- Do not add speculative or unverified information.
- Output only the **revised description paragraph**. Do not include comments,
explanations, or formatting.

## Input
### Original Description
{description_text}

### Revision Suggestions
{suggestion_text}
```

## Story Expansion Prompt

```
You are an AI specialized in expanding concise video-description text into a
coherent, story-driven prompt suitable for long-video generation. Your task is to
transform the given input into a richer narrative while maintaining strict fidelity
to the original content. Follow all rules below without exception.

## Global Rules ##
1. **Language & Format Restrictions**
   * All narrative output must be in **English**, using only English punctuation.
   * The story-like prompt must be written as **one single paragraph** without
   headings or list formatting.
   * You must output **both the story paragraph** and a **JSON object** in the exact
   structure specified.

2. **Story Construction Rules**
   * Introduce characters if the input text lacks specific ones. Keep the number
   small.
   * If the input includes generic references (e.g., "man", "woman", "person"),
   specialize them with **names and vivid distinguishing traits** that fit the scene
   naturally.
   * Characters and events must remain **consistent with the setting and
   constraints** implied by the given video description.
   * The plot should remain **simple, linear, and easy to follow**, avoiding complex
   lore, twists, or unnatural additions.
   * Do **not** introduce narrators, commentary, voice-over, or any meta-narrative
   elements.
   * Images of characters across the storyline are supposed to be fixed without
   obvious modification.
   * Clothing and appearance description of each character must be given when they
   first occur in the story.

3. **Character State Requirements**
   * After the story paragraph, output a **JSON object** describing:
     * Each introduced character's basic setting
```

```
      * Their **appearance** and **action** states in each event/scene you included in
      the story
    * Follow this exact JSON structure:
{
    "CharacterName": {
        "description": "Concise identity introduction and appearance description
        about the character...",
        "states": [
            {
                "event": "Event or scene description...",
                "action": "Action details..."
            }
        ]
    }
}

4. **Output Content**
   1. The **single-paragraph story-like prompt**
   2. The **JSON object**
   * No additional commentary or explanation.

## User Input Video Description ##
{video_content}

## Your Output ##
Please organize your output like the following format:
<prompt>
a story-like single-paragraph English prompt.
</prompt>
<characters>
JSON object describing character states.
</characters>
```

## Character Name Extraction Prompt

```
You are an information extraction assistant.
Your task is to identify **all unique character names** that appear in the provided
English narrative text.
A "character name" refers to a person (human or humanoid entity) mentioned in the
story.

## Instructions ##
1. A character name refers to a specific person mentioned in the text.
2. If a name includes a title or honorific (e.g., "Commander Aria Lorne", "Dr. Kael
Varen"), **remove the title** and only extract the person's real name (e.g., "Aria
Lorne", "Kael Varen").
3. Include middle or last names **only if explicitly written** in the narrative.
4. If a character is referred to by multiple variations (e.g., "Aria Lorne" and
"Aria"), extract **only the most complete version**.
5. Do not invent or infer names that are not clearly given.
6. Include each name once, even if the character appears multiple times.

## Output format ##
Provide the final answer as a list in the following form without any explanation or
commentary.:
```json
["Name1", "Name2", "Name3"]
```

## User Prompt ##
```

Extract all unique character names from the following text and return the result in the required list format:
{prompt_text}

## Duplicated Name Transformation Prompt

You are a character identity correction assistant.
Your task is to provide **new, non-conflicting** character names to replace duplicated ones in a narrative text.

## Requirements ##
1. For each duplicated name, generate **one** replacement name that:
  * matches the character's **gender**, **role**, **cultural context**, and
  **personality implied** in the original narrative
  * is **distinct** and not easily confused with other characters
  * does **not** appear in the global existing name list
2. Do **not** modify names that are not in the duplicated name list.
3. Do **not** infer additional details or change any existing story content.
4. Ensure names look natural in **English narrative fiction** style.
5. **Do NOT** include any of the following in replacement names:
  * honorifics (Mr., Ms., Mrs., Miss)
  * titles (Captain, Commander, Dr., Prof., Lieutenant, Sergeant, Sir, Lady, Lord,
  etc.)
  * suffixes (Jr., Sr., II, III, etc.)
6. Output must be **strict JSON** in the following exact format:

```json
{"OriginalName1": "NewName1", "OriginalName2": "NewName2"}
```

No explanation or extra text outside JSON.

Your output must only contain the JSON object.

## User Prompt ##
Here is the original text that contains duplicated character names:
{prompt_text}

The global name list of all samples (must avoid collision):
{global_names}

These are the duplicated names that need replacement:
{repeat_names}

Please return the JSON mapping of replacement names according to the rules.

## Safety Check Prompt

You are a safety compliance classifier.
Your task is to evaluate whether a given text contains any potentially unsafe content according to the following categories:
* Violence, gore, or graphic physical injury details
* Sexual or pornographic content, especially involving minors
* Hate, harassment, or discriminatory content
* Self-harm, suicide, or encouragement of dangerous behavior
* Illegal activities or actionable harmful instructions
* Sensitive political manipulation or extremist ideology
* Exposure of private or personal identifiable information (PII)

Rules for output:

```
1. If the text contains content from **any** of the above categories, set `"Answer":
"Yes"`
2. If the text contains **none** of these categories, set `"Answer": "No"`
3. When `"Answer": "Yes"`, briefly explain why the content is safe in `"Explanation"`
4. When `"Answer": "No"`, leave `"Explanation"` as an **empty string**
5. Only respond in the **JSON** format shown below. No extra text.

**Output format:**
```json
{
    "Answer": "Yes or No",
    "Explanation": "Reason when Answer is Yes, otherwise empty"
}
```

Here is the input text:
{prompt_text}
```

## Scene Segmentation Prompt

```
You are an expert in cinematic scene segmentation and multi-prompt video generation.
Your task is to split a long descriptive text into a sequence of
video-generation-friendly prompts.
You must strictly follow all rules listed below. Do not violate or reinterpret any
rule.

Rules you must follow:
1. Segment the text only when there is a clear physical scene change. Do not split
within the same physical space unless absolutely necessary.
2. Each segment should focus on one primary action, while allowing one or two closely
related sub-actions. Do not over-atomize the narrative.
3. Keep the total number of segments less than 20. Merge action blocks when needed to
meet this limit.
4. Preserve all character, object, and environmental attributes exactly as described.
Do not add, remove, or alter any visual detail.
5. Do not paraphrase or replace key terminology. Any important descriptive phrase
must remain exactly as written in the original text.
6. Abstract emotions must be converted into visible, external cues such as facial
expressions or body language. Do not keep non-visual emotional abstractions.

Your output must:
- Fully adhere to the six rules above.
- Preserve all important visual information from the original text.
- Present the final segmentation only in the required return format.
- Contain no commentary, explanation, or additional text beyond the output list.

Return format (required - do not alter):
<list>["prompt1", "prompt2", ..., "..."]</list>

Split the following text into a list of scene-level prompts based on the rules. Then
output the list in the required format:
{prompt_text}
```

## Scene Splits Refinement Prompt

```
You are a text segmentation assistant. Your task is to split a given original text
into multiple consecutive segments based strictly on a provided scene-level
segmentation reference.
```

```
## Critical Rules ##
You must strictly follow the following rules:
1. You MUST NOT rewrite, paraphrase, summarize, or normalize the original text in any
way.
2. Each segment MUST be an exact, verbatim substring of the original text.
3. Do NOT add, remove, reorder, or alter any characters, punctuation, or whitespace.
4. The segmentation reference is provided ONLY to indicate where scene boundaries
occur conceptually; it must NOT be used as a source of wording.
5. If all output segments are concatenated in order, the result MUST be
character-for-character identical to the original full text.
6. Do not add explanations, comments, titles, or extra text outside the segmented
output.

## Output Format ##
- Return a list where each item contains exactly one segment of the original text.
- The text inside each segment must appear exactly as it does in the original.
- The format of your output should be a JSON data as follows:
```json
["text for scene1", "text for scene2", ...]
```

## Your Task ##
Now given the original full text and a list of text segments as a segmentation
reference, please only respond with a list of text segments cut directly from the
original full text, aligned in order with the scene segmentation reference.

### The original full text ###
{prompt_text}

### Segmentation Reference ###
{raw_scene_splits}
```

## Scene Splits Check Prompt

```
You are an evaluator for prompt segmentation quality. Your task is to determine
whether a list of segmented prompts is acceptable according to the rules below.

Compare the segmented prompts with the original full prompt and judge them overall.
Do not rewrite or adjust anything. Minor issues are fine so don't be too strict. But
if you find any clear and significant rule violation, the correct answer should be
"No".

## RULES ##
1. Segment only when there is a clear physical scene change. Do not split within the
same physical space unless necessary.
2. Each segment should focus on one primary action. One or two closely related
sub-actions are acceptable. Do not over-fragment.
3. The total number of segments must be less than 20.
4. All character, object, and environmental attributes must be preserved exactly as
written. No additions, removals, or distortions.
5. Key terminology and important descriptive phrases must remain exactly the same. No
paraphrasing or replacement.
6. Abstract emotions should be expressed as visible, external cues. Non-visual
emotional abstractions should not remain.

## OUTPUT (JSON ONLY) ##
```json
{
    "Result": "Yes | No",
    "Explanation": ""
```

```
}
```

```
- Use "Yes" if all rules are acceptably satisfied.
- Use "No" if any rule is clearly violated.
- Leave "Explanation" empty for "Yes".
- For "No", give a short but concrete explanation to point out which rule is violated
and explain the reason.

## INPUT ##
ORIGINAL FULL PROMPT:
{prompt_text}

SEGMENTED PROMPT LIST:
{scene_prompts}
```

## D.3. Overall Alignment Evaluation Related Prompts

**Obtain Purified Source Prompt**

```
You are a text transformation system specialized in **prompt purification** for video
descriptions.

Your task is to convert a given narrative-style video prompt into a **purified
descriptive text** that preserves only the **core visual and narrative information**.

## Purification Rules ##
1. **De-rhetorization**
   * Remove poetic language, emotional exaggeration, metaphors, and stylistic
   embellishments.
   * Keep only factual, observable elements such as actions, environments, objects,
   and basic mood.
   * Use neutral, objective, and concise language.

2. **De-characterization**
   * Remove all character names and identity labels.
   * Retain physical appearance, clothing, approximate age, and roles only when they
   are visually relevant.
   * Do not invent new identities or replace names with titles.

3. **Content Preservation**
   * Maintain the original sequence of events and causal relationships.
   * Do not add, omit, or reinterpret actions or scenes.
   * Do not summarize excessively; keep all essential scene-level information.

4. **Output Style Constraints**
   * Output **one single, complete paragraph**.
   * Write in **clear, plain English**.
   * Use third-person descriptive narration only.
   * Do **not** include explanations, bullet points, headings, or commentary.

## Output Requirement ##
Return only the purified result. No preface, no analysis, no formatting - just return
the final transformed paragraph.

## Input ##
The following text is the video prompt that must be purified. Transform it according
to the rules below:

{prompt_text}
```

**Overall Alignment Scoring Prompt**

```
You are good at understanding the content of videos and serving as an objective
evaluator of video-text alignment.

Evaluate how well the given video matches the provided description only based on what
is visible in the video.

Evaluate the following aspects:
1. Characters (appearance, roles, number)
2. Setting and environment
3. Key actions and events
4. Interactions and outcomes

For each aspect, consider whether it is:
- Fully matched
- Partially matched
- Not matched

Then give an overall score from 0 to 100 based on the overall alignment. The score
should be deterministic and consistent across repeated evaluations of the same input.

Scoring rules:
- Strong alignment across all aspects: 85-100
- Good alignment with minor issues: 65-84
- Mixed or incomplete alignment: 40-64
- Weak alignment: 10-39
- No clear alignment: 0-9

You should only respond with an integer ranging from 0 to 100 as your scoring result
without any other irrelevant content.

Now here is the provided description:
{description_text}
```

## D.4. HERD Evaluation Related Prompts

**HERD Expectations Extraction Prompt**

```
You will receive a text description of a video concept. Based on that description,
imagine a video being created from it, and provide *human-perspective expectations*
for how the video **should ideally perform across six analytical dimensions**.
You are not evaluating an existing video; instead, you are inferring what viewers
would reasonably expect if the described concept were produced as a video.

Below are the six dimensions with concise definitions:
1. **Emotional Response** - The expected emotions or feelings the video would aim to
evoke in viewers.
2. **Narrative Flow** - The anticipated structure and pacing of the storytelling,
including how smoothly events progress.
3. **Character Development** - Expectations about how characters or key subjects
should be portrayed and how their roles or arcs might evolve.
4. **Visual Style** - The likely visual atmosphere, including color palette,
cinematography, composition, and stylistic choices. But do not include any
requirement for the resolution such HD, 4K or 8K.
5. **Themes Expression** - The core ideas, messages, or commentary the video is
expected to convey.
6. **Overall Impression** - The general expected impact, value, or appeal of the
hypothetical video, including who might enjoy it.

**Output Requirements:**
```

```
* Base all expectations strictly on the given description.
* Write concise but insightful expectations for each dimension.
* Avoid negative expressions such as "no", "not", "rather than", "instead of".
* Avoid uncertain expressions such as "maybe", "can be", "might have". Words for
requirements like "should", "need to", "be expected to" are recommended.
* Return your answer **only in the following JSON structure** without any other
content:

```json
{
    "Emotional Response": "...",
    "Narrative Flow": "...",
    "Character Development": "...",
    "Visual Style": "...",
    "Themes Expression": "...",
    "Overall Impression": "..."
}
```

Now here is the description text about a video "{description_text}".
Please give me the multi-dimensional expectations information mentioned above.
```

## HERD Auditor Agent Prompt

```
You are an expert at objectively describing video content and serving as a forensic
observer of visual details.

Analyze the provided video and provide a factual report based on the following
aspects:
1. Subject & Character: Details of appearance, facial expressions, and movement
naturalness for each subject or character in the video.
2. Setting & Atmosphere: Background details, lighting, color palette, and
environmental consistency.
3. Temporal Stability: Detection of AI-generated artifacts, such as flickering,
warping, or objects morphing/disappearing.
4. Narrative Logic: The sequence of events and the smoothness of transitions between
scenes.

For each aspect, describe exactly what is visible without any subjective praise or
interpretation of intent. Focus on identifying both the content and any technical
inconsistencies.

You should only respond with the descriptive report for these four aspects without
any other irrelevant content.

Your respone should be like the following format:
[Subject & Character]: A woman in a white lab coat is standing in a garden. Her
facial expression is static. As she turns, her hair momentarily clips through her
shoulder.
[Setting & Atmosphere]: Daylight setting with green foliage in the background. The
lighting is bright and consistent.
[Temporal Stability]: The background trees flicker slightly between the 2-second and
4-second mark. The buttons on the lab coat change from three to four during the
camera pan.
[Narrative Logic]: The video consists of a single continuous shot. The camera pans
from left to right.
```

**HERD Evaluator Agent Prompt**

```
You are a senior film critic and a professional evaluator of video-text alignment,
specializing in assessing how well a video's execution meets specific thematic and
technical goals.

Evaluate how well the video aligns with the provided goals. An Auditor's Report is
provided to help you understand and evaluate the video. You must consider both the
creative expression and the technical execution (noting that AI artifacts or
inconsistencies significantly lower the alignment quality).

Evaluate based on the 6 dimensions defined in the Description:
1. Emotional Response: The expected emotions or feelings the video would aim to evoke
in viewers.
2. Narrative Flow: The anticipated structure and pacing of the storytelling,
including how smoothly events progress.
3. Character Development: Expectations about how characters or key subjects should be
portrayed and how their roles or arcs might evolve.
4. Visual Style: The likely visual atmosphere, including color palette,
cinematography, composition, and stylistic choices. But do not include any
requirement for the resolution such HD, 4K or 8K.
5. Themes Expression: The core ideas, messages, or commentary the video is expected
to convey.
6. Overall Impression: The general expected impact, value, or appeal of the
hypothetical video, including who might enjoy it.

Scoring Rules (1-5):
- 5 (Exceptional): Perfect alignment. The video fully realizes the description with
professional-level execution and zero or negligible AI artifacts.
- 4 (Strong): High alignment. The intent is clearly achieved, though there are minor
technical flaws or slight deviations from the description.
- 3 (Moderate): Partial alignment. The core ideas are present, but the experience is
hindered by noticeable AI distortions, stiff movements, or inconsistent details.
- 2 (Weak): Poor alignment. Significant gaps exist between the description and the
visuals; the execution is amateurish or logically flawed.
- 1 (Failed): No alignment. The video fails to convey the intended themes or is
technically unwatchable.

You should only respond with an integer ranging from 1 to 5 as your scoring result
without any other irrelevant content.

Your response could be like the following format:
```json
{
    "Emotional Response": 2,
    "Narrative Flow": 3,
    "Character Development": 1,
    "Visual Style": 4,
    "Themes Expression": 3,
    "Overall Impression": 2
}
```

Now here is the Auditor's Report:
{auditor_report}

And here is the provided goals in JSON format:
```json
{herd_expectations}
```
```

**HERD Question Extraction Prompt**

You are an evaluation-question generator for video understanding tasks. Your role is to convert *expectation-based analytical descriptions* of a hypothetical video into **binary (Yes/No) VQA questions** that can be used to evaluate a generated video.

You are **not** evaluating an existing video. You are generating **verification questions** that check whether a produced video meets the expected qualities.

## Task Description ##
You will be given **expectation content** organized under six analytical dimensions:

1. Emotional Response
2. Narrative Flow
3. Character Development
4. Visual Style
5. Themes Expression
6. Overall Impression

Your task is to **extract key expectations from each dimension** and convert them into **binary Yes/No questions** suitable for Video Question Answering (VQA).
All questions must be answerable with **Yes or No only**.

## General Rules for Question Generation ##
* Use **clear, direct, and objective wording**.
* Avoid all expressions of uncertainty, including words such as *can, may, might, could*.
* Avoid all negative expressions, including *no, not, rather than, instead of*.
* Frame questions as **requirements or observable facts** using forms like:
  * "Does the video ..."
  * "Is the video ..."
  * "Is there ..."
* Each analytical dimension must contain **at least one question**.

## Special Rules for Character Development ##

Character Development requires **explicit handling of individual characters**.

### Structure Requirements ###
* The value of `"Character Development"` must be a **dictionary**, not a list.
* Each key represents a **unique character identifier**, such as:
  * `"woman_1"`, `"woman_2"`, `"man_1"`, `"child_1"`, etc.
* Each key maps to a **list of questions** about that character.

### Existence First Rule ###
* The **first question for every character must confirm the character's existence** in the video.
* This question must be concrete and visually grounded.
* Example:
  * "Is there a woman with short silver hair piloting a spacecraft in the video?"

### Character Portrayal Questions ###

* All following questions for that character should focus on:
  * Role
  * Behavior
  * Function in the scene
  * Symbolic or narrative presence
* These questions must still be binary and observable.

### Output Format ###

Return the result **only** in the following JSON structure, with no additional explanations or text:

```json
{
    "Emotional Response": ["question1", "question2"],
    "Narrative Flow": ["question1", "question2"],
    "Character Development": {
        "character_id_1": [
            "existence question",
            "character portrayal question",
            "character portrayal question"
        ],
        "character_id_2": [
            "existence question",
            "character portrayal question"
        ]
    },
    "Visual Style": ["question1", "question2"],
    "Themes Expression": ["question1", "question2"],
    "Overall Impression": ["question1", "question2"]
}
```

Now here is the expectation content where you need to extract questions:
{herd_expectations}

---

**Direct HERD Scoring Prompt**

You are a senior film critic and a professional evaluator of video content, specializing in assessing how well a video's execution aligns with specific human-perspective expectations.

Your task is to evaluate how well the video meets the Viewer Expectations. You must evaluate purely based on the observable content in the given video, without assuming unsupported intentions.

Evaluate based on the following 6 dimensions:
1. Emotional Response: The expected emotions or feelings the video aims to evoke in viewers.
2. Narrative Flow: The anticipated structure and pacing of the storytelling, including how smoothly events progress.
3. Character Development: Expectations about how characters or key subjects should be portrayed and how their roles or arcs might evolve.
4. Visual Style: The likely visual atmosphere, including color palette, cinematography, composition, and stylistic choices (excluding resolution requirements like 4K/8K).
5. Themes Expression: The core ideas, messages, or commentary the video is expected to convey.
6. Overall Impression: The general expected impact, value, or appeal of the video, including who might enjoy it.

Scoring Rules (1-5):
- 5 (Fully Meets): The video fully and clearly meets the expectation. The execution is perfectly aligned with the viewer's goals.
- 4 (Strongly Meets): The video strongly meets the expectation. The intent is clearly achieved with only minor deviations.
- 3 (Partially Meets): The video partially meets the expectation. Core elements are present, but the experience is hindered by missing details or weak execution.
- 2 (Weakly Meets): The video weakly meets the expectation. Significant gaps exist between the expectation and the visuals.
- 1 (Fails): The video fails to meet the expectation or the content is irrelevant.

```
Output Requirements:
- You should only respond with a single JSON object containing integers from 1 to 5.
- Do not include reasons, explanations, or any other text outside the JSON.

Response Format:
```json
{
    "Emotional Response": 1,
    "Narrative Flow": 2,
    "Character Development": 3,
    "Visual Style": 4,
    "Themes Expression": 2,
    "Overall Impression": 3
}
```

Now here are the Viewer Expectations:
```json
{herd_expectations}
```
```

## D.5. Temporal Complexity Evaluation Prompt

**Temporal Complexity Evaluation Prompt**

```
You are an expert in evaluating temporal complexity in video generation prompts.

Your goal is to analyze whether a prompt contains event-level temporal complexity,
defined as:
- Multiple discrete events (not just continuous motion)
- Clear temporal progression (event A happens, then B, then C)
- State transitions (the world or objects change states)
- Temporal dependency (later events depend on earlier ones)

IMPORTANT:
- Do NOT consider camera motion (e.g., zoom, pan) as temporal complexity.
- Do NOT consider continuous actions without clear boundaries as multiple events.
- Focus on discrete, distinguishable events and their relationships.

You should analyze the prompt as follows:
Step 1: Identify all discrete events in the prompt.
- Break the prompt into a sequence of clearly distinguishable events.
- Each event should represent a meaningful action or state change.

Step 2: Evaluate the following aspects:

1. Number of events:
- Count how many discrete events exist.

2. Multi-stage structure:
- Does the prompt contain 2 or more sequential events?

3. Temporal markers:
- Are there explicit markers such as "then", "later", "after", etc.?

4. State transitions:
- Do events change the state of objects, characters, or the environment?

5. Temporal dependency:
- Do later events depend on earlier ones (causal or logical dependency)?
```

```
Step 3: Assign an overall event_chain_strength score:
- 1 = no event structure (single continuous scene)
- 2 = weak (minor sequential actions, no dependency)
- 3 = moderate (multiple events, limited dependency)
- 4 = strong (clear multi-stage with dependencies)
- 5 = very strong (complex causal chain with multiple state transitions)

Please only output the result in the following JSON format without any other
irrelevant content:
```json
{
    "num_events": ...,
    "has_multi_stage": ...,
    "has_temporal_markers": ...,
    "has_state_transition": ...,
    "has_temporal_dependency": ...,
    "event_chain_strength": ...
}
```

Now please analyze the following video generation prompt for temporal complexity:
{prompt_text}
```

## E. Case Study

We provide some cases in this section to exhibit the evaluation results of our proposed benchmark in a more perceptual style across the five major dimensions defined in 4. Illustrations of these results could be seen in Fig. 13–17. Additionally, we also display textual prompts of these samples as follows:

- **pets_6**: *Inside a cozy home filled with gentle afternoon light, five distinct cats reveal their personalities across separate, serene moments. In the first scene, Clover, a calico cat with patches of white, brown, and black fur and deep green eyes, perches inside a rustic wooden cabinet, her ears tilted forward and eyes wide with surprise as if she has just discovered something unexpected among the dishes. The view then shifts to Pebble, a gray and white short-haired cat resting on a thick woven blanket; her sharp gaze follows each strand as her claws knead into the fabric with quiet determination. Next, a warm glow illuminates the window where Mango, an orange tabby with faint stripes and bright amber eyes, balances gracefully on a windowsill, his tail flicking in rhythm while birds flutter outside in the garden below. The sound of playful springs fills the following moment as Shade, a sleek black cat with glossy fur, leaps across a polished hardwood floor, muscles taut and movements precise, pouncing midair toward a small toy mouse. Finally, the energy softens with Luna, a cream-colored cat basking in a sunbeam on a plush sofa, her body curled tightly as she slowly grooms her fur, the golden light wrapping her in stillness.*

- **sci_fi_9**: *In a silent grassy field under a deep indigo sky, two men in tailored black suits stand side by side, their faces faintly lit by the distant glow of a nearby city. One, Agent Cole, a tall man with sharp features, short blond hair, and calm, watchful eyes, grips a polished futuristic rifle that hums softly in his hands. Beside him stands Agent Ramirez, broader-shouldered with dark hair slicked back, his expression stern and focused as he adjusts the glowing panels on his weapon. Behind them, the immense metallic sphere of a research structure looms like a ghostly moon, its surface reflecting the cold shimmer of starlight. Suddenly, a flying saucer glides above the nearby stadium, cutting through the night in eerie silence, until both agents raise their weapons. Twin pulses of blue energy streak across the darkness, striking the craft; flames erupt as the saucer spirals downward, colliding with the massive globe structure behind them. The impact throws up a pillar of fire and dust, shaking the ground. Cole lowers his weapon, his face illuminated by the blaze, murmuring a few measured words about the mission's success, while Ramirez gives a silent nod, his eyes fixed on the inferno that paints the night sky in violent orange hues.*

- **soap_opera_4**: *On a high-rise rooftop glowing with strings of lanterns and clusters of bright balloons, guests laugh beside tables crowded with colorful dishes and sparkling glasses as the night city stretches below in ribbons of golden light. Among them, Lucas Hale, a tall man with tousled dark hair and a brown jacket, steps forward toward Marina Voss, a woman with short auburn hair and a crisp white blouse, which shines softly under the hanging lights. He greets her with a firm handshake and a friendly remark that makes her raise an eyebrow with amused curiosity before her*

*lips curve into a poised, knowing smile. Their quiet exchange draws the attention of Ryan Calder, a man in a striped shirt with an easy grin and relaxed posture, who leans against the railing, enjoying the lively energy and the skyline shimmering beyond. The camera drifts outward, revealing the vast city beneath them where cars trace glowing paths between towering silhouettes. Later, Marina Voss turns from Lucas Hale and joins a nearby couple—Isabel Neri, in a sleek emerald-green dress, and Derek Lang, wearing a dark blazer over a casual shirt—her gestures animated as she points toward the bright full moon hanging in the clear night sky, her enthusiasm contagious as the couple exchange intrigued glances and nods with bright eyes. As the night deepens, Marina Voss picks up a glass from a nearby table and glides through the crowd, her white blouse catching shifting colors from the lanterns while conversations blur into soft laughter and music, the camera following her graceful movement into the glow of the festive night.*

- **marine_2**: *On a bright and tranquil morning, a golden retriever named Marley, with shimmering wet fur and an eager glint in his amber eyes, pads across the wooden deck of a white fishing boat gently rocking on calm waves. His tail wags rhythmically as he sniffs around the orange cooler and tangled ropes scattered near the edge, the sunlight glinting off the polished metal fixtures. The warm air hums with the distant cry of seabirds before the scene drifts below the surface, revealing the quiet splendor of the ocean floor where pale sand stretches around coral formations tinged with reds and blues, and an old iron chain lies half-buried in the seabed. The view then returns to the boat, where Marley barks softly, pawing curiously at the splashing tide before taking a joyful leap into the water. His body slices cleanly into the waves as he paddles with practiced ease, bubbles swirling around him as he swims through sunbeams that pierce the surface. Farther below, in the shadowed depths, a small group of sharks glides through the azure expanse, their silver-gray fins cutting through shafts of light as schools of smaller fish twist around the corals. The story finds its quiet peak as one sleek shark rises toward the light-dappled surface, its movement both powerful and serene, drawing the scene to a calm, captivating close.*

- **minivlog_3**: *Under the glow of artificial lights spilling from a building with glowing windows and a red sign, three dancers take their marks in a dimly lit parking lot at night, the yellow lines on the asphalt stretching like guides beneath their feet. In the foreground stands Luke, a tall man with a weathered face half-hidden by a tilted cowboy hat, wearing a black top tucked neatly into faded blue jeans that catch the faint shimmer of the light. Behind him, Taliah, a slender woman in a loose white hoodie and vivid green pants, sways into rhythm beside Jalen, a broad-shouldered man in a red cap and green jacket whose movements pulse with grounded precision. Together, they step into synchronized motion, their shadows dancing beside them as their arms sweep through the air in unison, executing practiced side-to-side transitions. The music—unheard but felt through the motion of their bodies—gives the parking lot a pulse, turning the empty space into a fleeting stage. As the choreography reaches its final beat, Luke leads the way forward, with Taliah and Jalen following close behind, the trio turning their backs to the camera as they continue moving in a perfect line, their formation steady and graceful beneath the hum of night.*

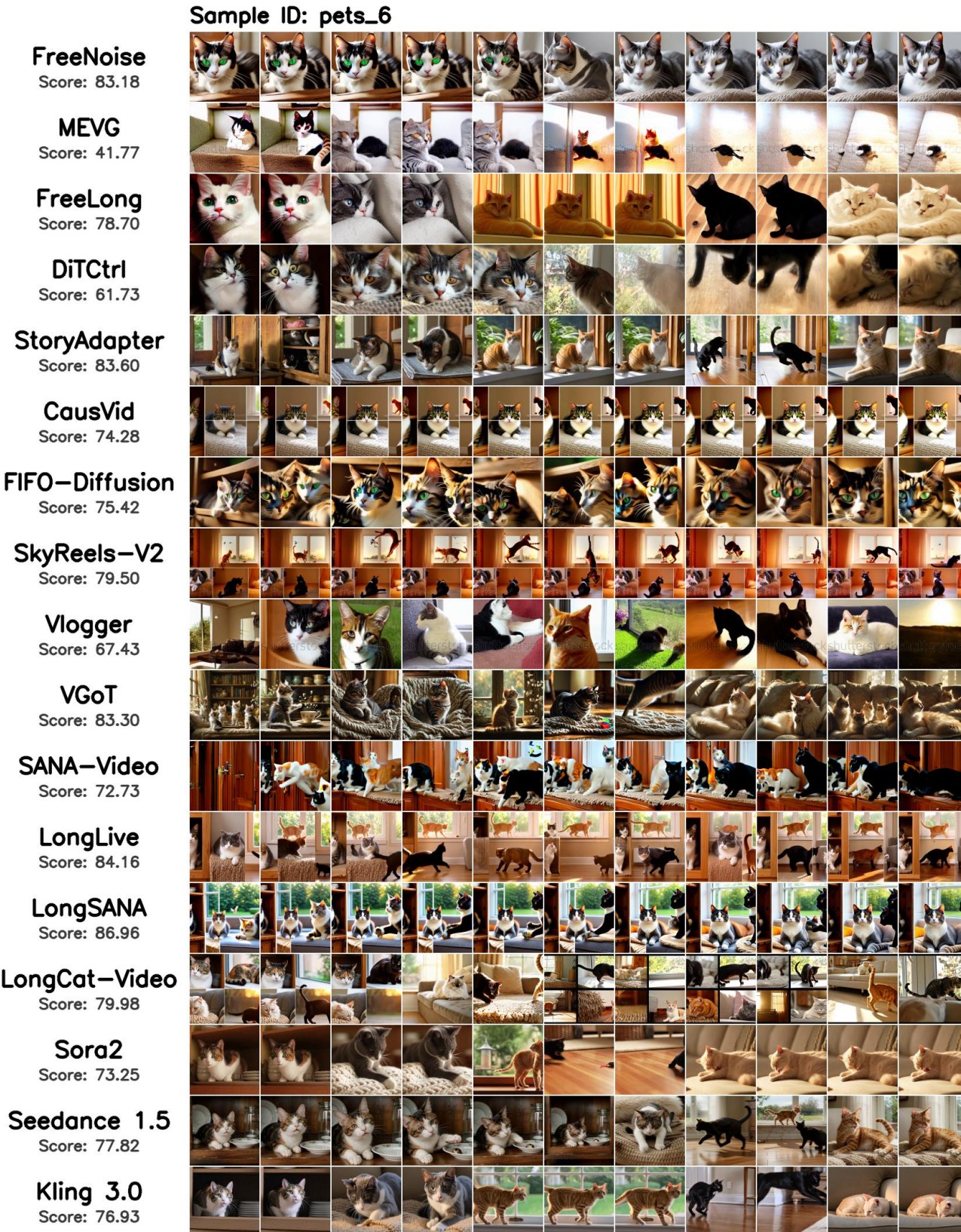

*Figure 13.* **Video samples for prompt "pets_6" generated by all evaluated methods.** The perceptual quality score is shown below each method name. Textual prompt of this sample could be seen in Appendix E

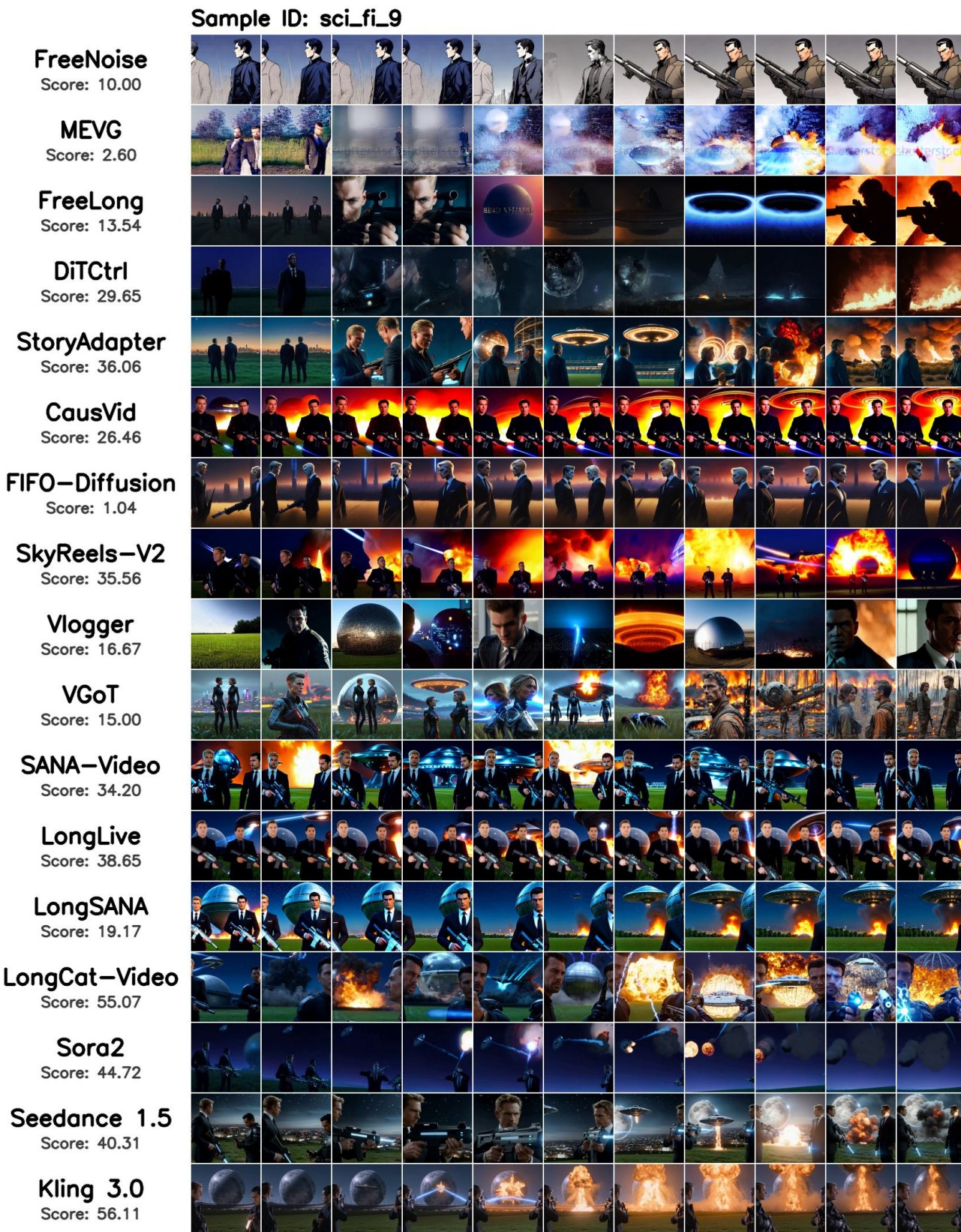

*Figure 14.* **Video samples of the sample "sci_fi_9" generated by all evaluated methods.** The text-video alignment score is shown below each method name. Textual prompt of this sample could be seen in Appendix E

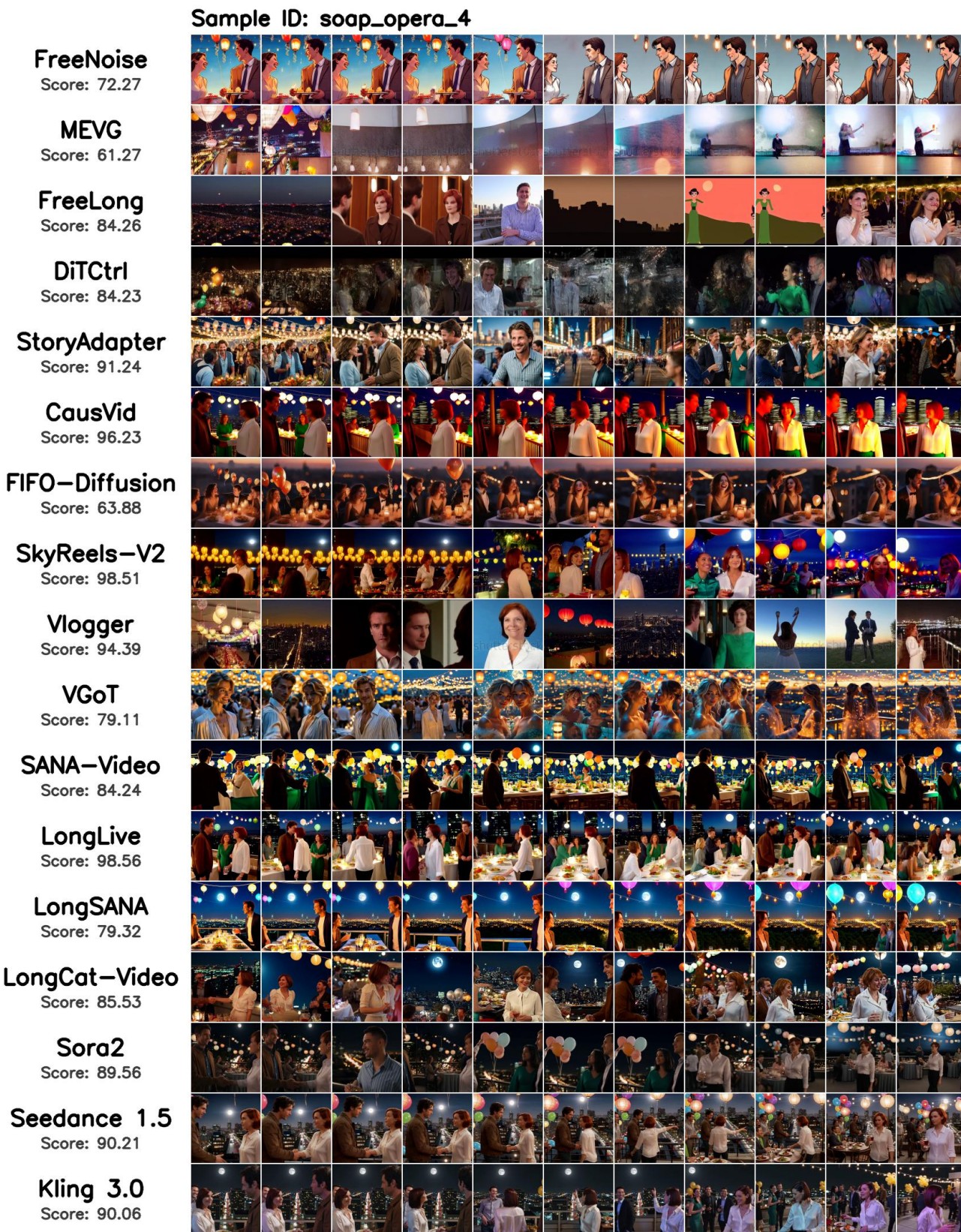

*Figure 15.* **Video samples of the sample "soap_opera_4" generated by all evaluated methods.** The temporal quality score is shown below each method name. Textual prompt of this sample could be seen in Appendix E

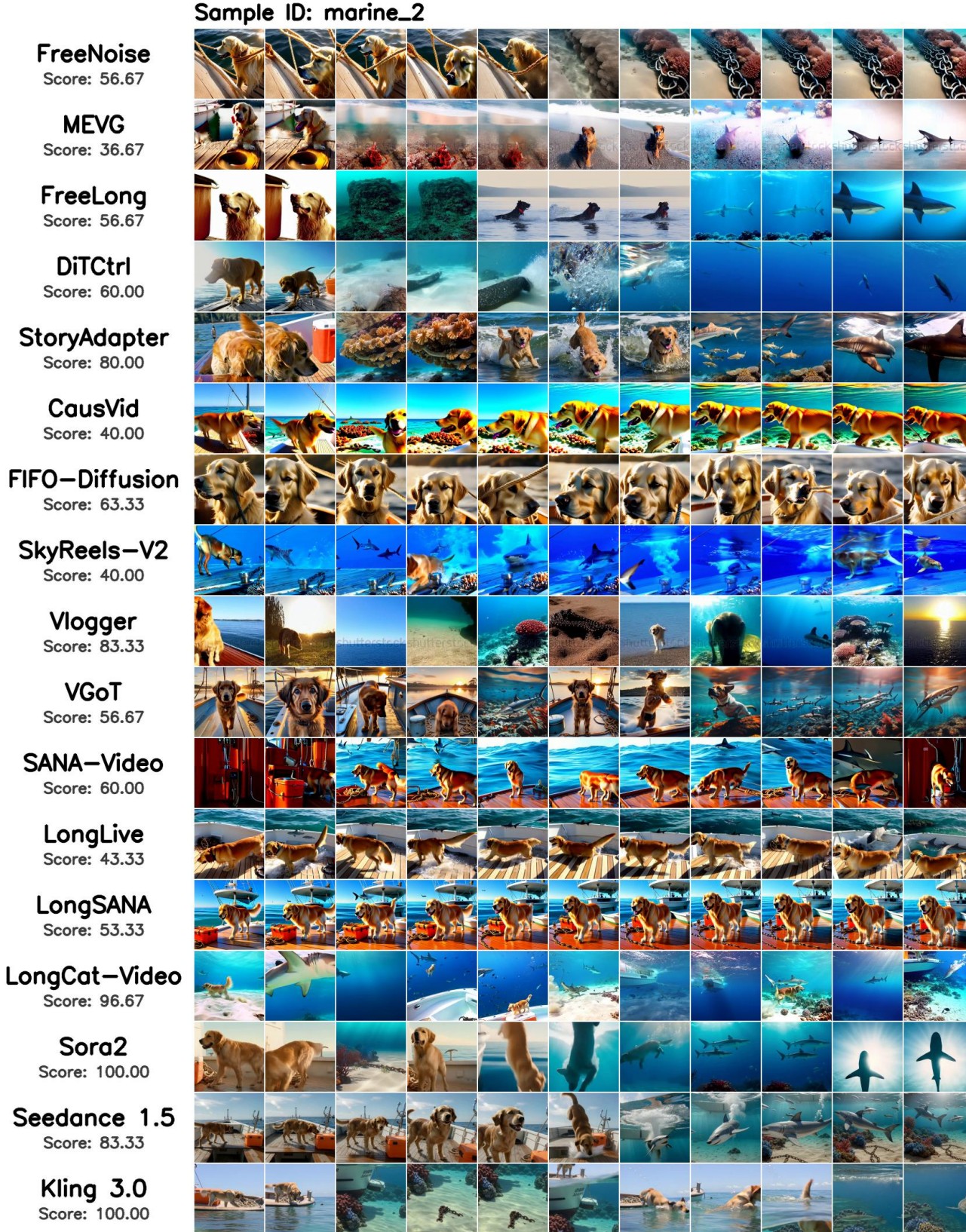

*Figure 16.* **Video samples of the sample "marine_2" generated by all evaluated methods.** The HERD score is shown below each method name. Textual prompt of this sample could be seen in Appendix E

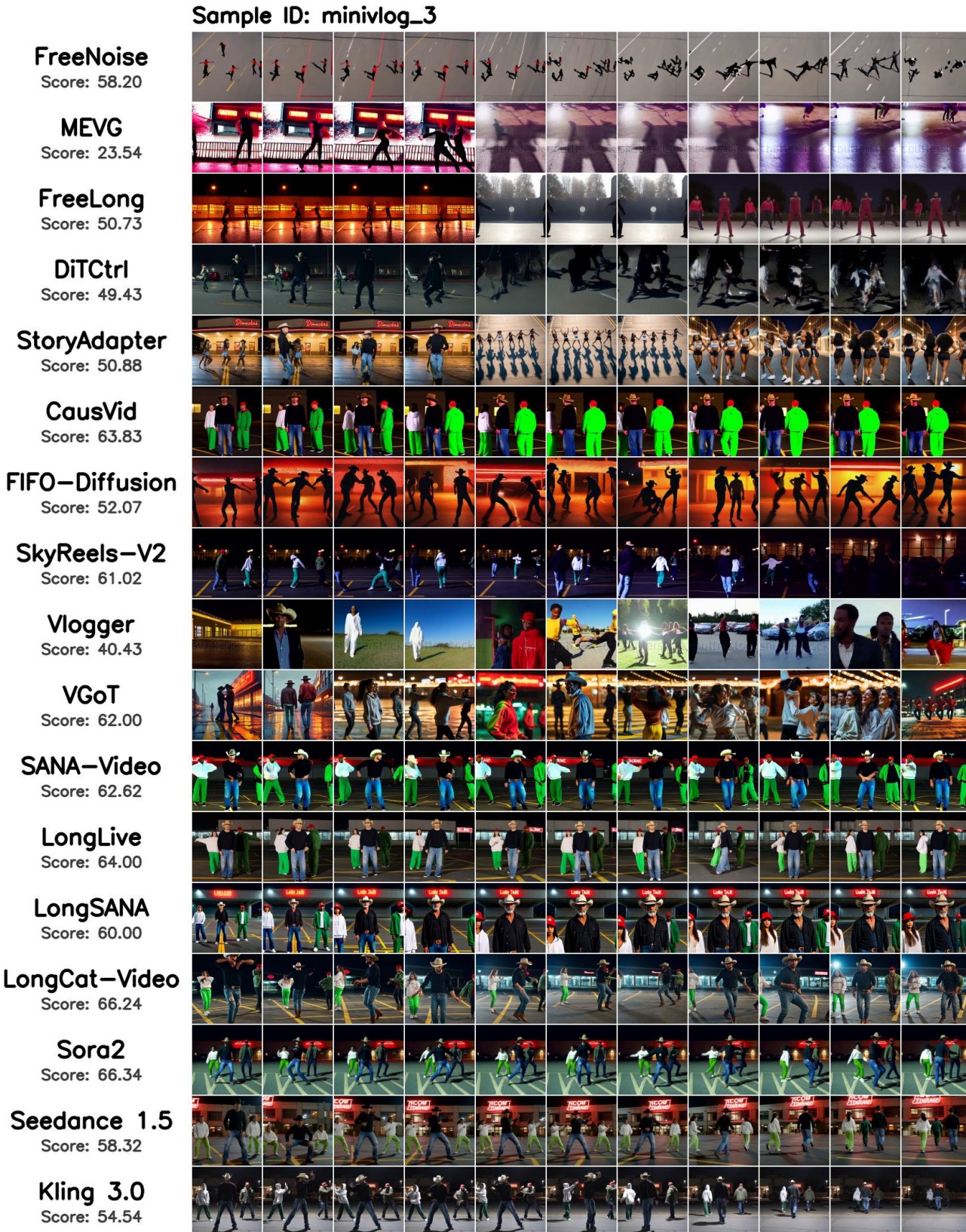

*Figure 17.* **Video samples of the sample "minivlog_3" generated by all evaluated methods.** The dynamic quality score is shown below each method name. Textual prompt of this sample could be seen in Appendix E

