# OpenReview forum: "LoCoT2V-Bench: Benchmarking Long-Form and Complex Text-to-Video Generation"
_ICML.cc/2026/Conference — ICML 2026 regular_

### Official Review · Reviewer_bsLe · 2026-02-16

**Soundness:** 3
**Presentation:** 3
**Significance:** 3
**Originality:** 3
**Overall Recommendation:** 4
**Confidence:** 4

**Summary:**

This paper introduces LoCoT2V-Bench and LoCoT2V-Eval, a benchmark and evaluation suite for long-form, multi-scene text-to-video generation under complex prompts. The benchmark provides hierarchical prompt metadata and evaluates a range of baselines across perceptual quality, text-video alignment, temporal consistency, dynamics, and a human-expectation metric.

**Compliance With Llm Reviewing Policy:**

Affirmed.

**Final Justification:**

The authors have now included experiments with strong baseline models on the full benchmark suite, so I have decided to raise my score to weak accept. However, the work initially lacked many baselines as well as a clear definition and explanation of the method’s motivation (a concern similar to that raised by reviewer Nn4N), which still makes this a borderline case. The authors should try to address these issues as thoroughly as possible in the revised version.

**Key Questions For Authors:**

Please see weaknesses.

**Limitations:**

yes

**Strengths And Weaknesses:**

**Strengths**

1. The paper targets an important and under-explored setting: long-form, multi-scene text-to-video generation under complex prompts.

2. The writing is clear, and the construction of the core evaluation metrics is easy to follow.

**Weaknesses**

1. The parameter counts of the evaluated models are unknown (e.g., it is unclear whether SkyReels-V2 uses the 1.3B or 14B variant). Therefore, the benchmark’s reported performance ceiling may be misleading.

2. Several strong baselines are not included, such as the open-source model LongCat-Video [1], which can also generate long videos. In addition, it would be beneficial to include some commercial models as references (i.e., Kling, Seedance), so that readers can better gauge the difficulty of the task and the remaining challenges.

3. Based on the appendix cases, the current prompt suite does not seem to emphasize temporal complexity. For example, soap_opera_4 mainly involves a camera move from far to close, without much complex temporal structure. It focuses more on fine-grained spatial details than on temporally intricate content.

4. I suggest the authors include prompts similar to those in StoryEval-Bench [2] that involve multiple action transitions or require complex temporal reasoning, as these are key evaluation scenarios for long video generation. My main concern is that if the temporal content in the prompts is too simple and can be adequately expressed with short clips, then it undermines the purpose of evaluating long video generation in the first place.

5. Some implementation details are unclear. The authors do not specify whether the MLLM used in the VQA-based metrics is commercial or open-source, which is important for reproducibility.

[1] LongCat-Video Technical Report

[2] Is Your World Simulator a Good Story Presenter? A Consecutive Events-Based Benchmark for Future Long Video Generation

---

> ### Author Rebuttal · Authors · 2026-03-31
>
> We greatly appreciate the reviewer's insightful observations and constructive critiques. A point-by-point response is provided below.
>
> **[W1: Lack of Clarification on the Scale of Evaluated Models]**
>
> **R1**:  We appreciate the reviewer's suggestion for clarity. To accurately represent the **performance ceiling** of evaluated methods, we consistently selected the most capable variants available. Specifically, for **SkyReels-V2**, we utilized the **14B** variant with **540P** resolution. We will complement corresponding explanation in the revised version.
>
> **[W2: Lack of Evaluation on Strong Baselines and Commercial Methods]**
>
> **R2**: We appreciate the reviewer's suggestion to include these contemporary models, which helps provide a clearer picture of the task's difficulty and the state of the art.
>
> * **Inclusion of LongCat-Video**: We thank the reviewer for suggesting **LongCat-Video** as an important open-source baseline. Due to its high generation cost (≈24 hours for 5 videos), evaluating all 234 prompts is infeasible within the rebuttal budget. We therefore evaluate it on a representative subset of 50 prompts and report the results below. Full evaluation will be included in the revised version.
>
>   |Model|PQ|TVA-OA|TVA-FGA|TVA-Avg.|TQ-CC|TQ-BC|TQ-WE|TQ-Avg.|HERD|DQ|Avg.|
>   | - | :-: | :-: | :-: | :-: | :-: | :-: | :-: | :-: | :-: | :-: | :-: |
>   |LongCat-Video|78.33|67.68|48.88|58.28|52.18|98.45|95.32|81.98|87.07|60.01|73.13|
>
> * **Commercial Models (Kling, Seedance)**: We agree that including commercial models would provide valuable reference points. We have conducted a **small-subset evaluation on 4 representative commercial models**, and report the results in our response **R2** to **Reviewer pxYm**, along with detailed discussion on scope and limitations.
>
> **[W3/W4: Lack of Emphasis on Temporal Complexity]**
>
> **R3**: We highly recognize the reviewer's emphasis on temporal complexity as an important aspect of prompt design. While "temporal complexity" lacks a universal formal definition, we interpret it as the **intricacy of narrative progression, state transitions and causal dependencies** beyond simple camera motion according to the reviewer's description and will address your concern as follows:
>
> * **Case analysis**: We respectfully clarify that this case involves more than a simple camera movement. The prompt includes **multiple temporally separated segments** (e.g., the transition marked by *"Later"*) and requires maintaining **long-term character consistency**, **scene coherence**, and **logical progression of interactions** across segments. It also contains **multi-step interaction dynamics** (e.g., greeting → reaction → expression shift) and **shifts of attention across multiple characters**, which introduce non-trivial temporal dependencies.
>
> * **Dataset-level analysis**: To better ground this concept, we carefully design evaluation prompt and leverage GPT-5 to assess how the prompts of StoryEval-Bench and our LoCoT2V-Bench achieve this kind of temporal complexity from these aspects:
>
>   * **multi-stage structures**: Does the prompt contain 2 or more sequential events?
>   * **state transitions**: Do events change the state of objects, characters, or the environment?
>   * **temporal dependencies**: Do later events depend on earlier ones (causal or logical dependency)?
>
>   We provide the evaluation results as follows and conclude that our benchmark already contains **comparable temporal complexity**. Moreover, we would like to complement relevant comparison results for reflecting temporal complexity in the prompts of each benchmark and also detail our setting on this point. Details of this evaluation experiments like prompt setting could be seen on [this site](https://anonymous.4open.science/r/LoCoT2V-Bench-1518/).
>
>   | Benchmark | Multi-stage Ratio | State Transition Ratio | Temporal Dependency Ratio | Avg. Ratio |
>   | - | :-: | :-: | :-: | :-: |
>   | StoryEval-Bench | 1.00 |  1.00 | 1.00 | 1.00 |
>   | **LoCoT2V-Bench (ours.)** | 0.98  | 0.97 | 0.93 | 0.96 |
>
> **[W5: Lack of Clarification on Certain Implementation Details]**
>
> **R4**: We thank the reviewer for pointing this out. The MLLM used in our VQA-based metrics is the open-source **Qwen3-VL-8B-Instruct**, which demonstrates strong video understanding capabilities across various benchmarks. We adopt it as a practical trade-off between evaluation quality and cost, as our setting involves long videos with high token consumption, making commercial APIs prohibitively expensive. We will clarify this in our revised version.

---

> > ### Author Rebuttal · Reviewer_bsLe · 2026-04-03
> >
> > Thank you for the rebuttal. I appreciate the authors’ analysis of temporal complexity. However, the comparison against commercial models and strong baselines is limited to a small set of examples, which may introduce potential bias. While reporting results on the full benchmark suite would be important, these results are not available in the current version. Therefore, I am inclined to maintain my score, but remain neutral on acceptance.

---

> > > ### Author Response · Authors · 2026-04-06
> > >
> > > Dear Reviewer bsLe,
> > >
> > > Thank you for your constructive feedback regarding the limited comparison and potential bias. To address this concern, according to your suggestion, we did our best to conduct the evaluations on the full benchmark suite over the past few days.
> > >
> > > Specifically, we showed results of previously mentioned strong baselines and commercial methods **including LongCat-Video, Sora2, Seedance 1.5 pro and Kling 3.0**. Regarding Seedance 2.0, **since its API is currently not publicly accessible**, we are precluded from including its results in this response. We observed that the full-benchmark evaluation yields more stable (and slightly lower) scores across dimensions, alleviating the concern of potential inflation from limited samples, while preserving the relative performance trends across methods. Meanwhile, it further highlights the strong performance of commercial models and strong baselines, particularly in prompt adherence (perform much better in Text-Video Alignment).
> > >
> > > |Model|PQ|TVA-OA|TVA-FGA|TVA-Avg.|TQ-CC|TQ-BC|TQ-WE|TQ-Avg.|HERD |DQ|Avg.|
> > > |-|:-:|:-:|:-:|:-:|:-:|:-:|:-:|:-:|:-:|:-:|:-:|
> > > |LongCat-Video|**77.84**|64.52|49.61|57.06|42.71|98.28|94.88|78.62|85.13|59.28|71.59|
> > > |Sora2|66.59|69.64|54.09|61.87|**45.40**|99.10|98.41|**80.97**|86.42|**64.78**|**72.13**|
> > > |Seedance 1.5 pro|70.71|67.58|47.17|57.38|38.25|**99.13**|97.72|78.37|86.41|57.21|70.02|
> > > |Kling 3.0|70.26|**73.08**|**56.94**|**65.01**|36.97|98.96|**99.73**|78.55|**87.47**|56.16|71.49|
> > >
> > > We believe that these results will significantly strengthen our work, and we will incorporate the suggested additions in the revised version. Meanwhile, we are committed to opening all datasets and source code so that other researchers can easily reproduce our work.
> > >
> > > Thank you again for your valuable and insightful suggestions.

---

### Official Review · Reviewer_pxYm · 2026-02-27

**Soundness:** 3
**Presentation:** 4
**Significance:** 3
**Originality:** 3
**Overall Recommendation:** 5
**Confidence:** 4

**Summary:**

This paper provides a new benchmark for evaluating control of complex text in the current long video generation field. It establishes metrics to assess video quality from five aspects: Perceptual Quality, Text-Video Alignment, Temporal Quality, Dynamic Quality, and HERD, and evaluates recent methods.

**Compliance With Llm Reviewing Policy:**

Affirmed.

**Final Justification:**

The paper is clearly written and highly readable. Additionally, the authors have addressed my concern regarding the limited sample size in the rebuttal. Given the high complexity of the long videos, I consider 234 samples to be sufficient. I suggest that the authors consider further enriching the dataset upon open-sourcing.

**Key Questions For Authors:**

1) Why do the models evaluated in the article seem to not include mainstream T2V models such as Sora and Wan?

2) Is 234 samples too few? On average, there are even fewer samples under each topic. Can they represent enough scenarios?

**Limitations:**

yes

**Strengths And Weaknesses:**

Strengths: The paper provides a very comprehensive discussion and introduces a new evaluation benchmark for the video generation field, featuring complex prompts and multi-scene transitions. By designing comprehensive evaluation metrics across multiple dimensions and conducting experimental analyses on how different types of models respond to each metric, this work presents a novel and complete contribution.
﻿
Weaknesses: Although the prompts are significantly enriched and the metric design is comprehensive, the scale of the dataset is relatively small, consisting of only 234 samples.

---

> ### Author Rebuttal · Authors · 2026-03-31
>
> We appreciate the reviewer's valuable input and the effort devoted to assessing our submission. We respond to each point in detail below.
>
> **[W1/Q2: Limited Benchmark Scale]**
>
> **R1**: We really understand the reviewer's concerns about the scale of our benchmark. While 234 samples might appear small, our LoCoT2V-Bench prioritizes **evaluation density** and **prompt complexity**, distinguishing it from "large-scale but simple" benchmarks:
>
> * **Quality over Quantity:** Our prompts average **248.85 words** (Complexity: 8.70), far exceeding VBench-Long (7.64 words, 2.54 complexity).
> * **Granular Evaluation Points:** With **6.0 scenes/sample** (avg. duration 46.7s), the benchmark provides **>1,400 scene-level evaluation points**. This equals the semantic depth of 1,400+ short-form samples.
> * **Practicality:** High-fidelity LVG is resource-intensive; models like SkyReels-V2 require **2-4 hours per video** (2$\times$H20). Multi-scene LVG also involve frequent commercial LLM API calls even for every single video, which absolutely render high economic costs during evaluation. A massive scale would make the benchmark inaccessible to most researchers.
> * **Diversity:** Samples are evenly distributed across **18 diverse themes**, ensuring broad coverage of real-world and imaginative scenarios.
>
> **[Q1: Exclusion of Mainstream Models (Sora/Wan)]**
>
> **R2**: We fully understand the reviewer's confusion about why we don't introduce mainstream models, especially the commercial models with excellent performance in our evaluation. The exclusion of certain commercial/API-based models is due to **technical mismatch** and **cost constraints**:
>
> * **Economic Cost:** Commercial APIs cost a lot for generating per video especially longer video (e.g., one 12s video generated by Sora 2 would cost ~$2). A full evaluation (234 prompts) would cost thousands of dollars, which is prohibitive for academic benchmarking. However, we are open to adding a **small-subset case study** of Sora/Wan in the revised version to provide a qualitative reference.
>
> * **Temporal Duration Mismatch**: Our benchmark focuses on *true* long-form content. The basic duration statistics of our collected source videos are as follows. We expect their corresponding generation results have similar duration while Most current commercial APIs and mainstream models (like Wan2.2) are optimized for 5–12s clips, which does not meet our requirement.
>
>   |   | Min. | Max. | Avg. |
>   | - | :-: | :-: | :-: |
>   | Source Video Duration (s) | 29.75 | 64.24 | 46.74 |
>
> * **Proxy via Story Visualization**: To evaluate the capabilities of mainstream architectures like **Wan2.2**, we employed a cascaded **StoryAdapter + Wan2.2-I2V** approach. We believe the generation capacity of T2V and I2V within the same model family is comparable. This hybrid approach serves as a reference for multi-scene long-video generation, though it is more dependent on the story planning model.
>
> * **Results on Subset to Demonstrate**: We also extracted 10 representative prompts from our benchmark to evaluate state-of-the-art commercial models. While limited by budget, these results provide a crucial qualitative and quantitative reference for the community—Current long-video generation models achieve robust perceptual quality and environmental stability, but they exhibit a significant performance gap in fine-grained instruction following and subject identity preservation. These findings highlight that achieving precise semantic alignment and long-term character consistency remains the primary challenge for the field.
>
>   | Model | Duration |  PQ  | TVA-OA | TVA-FGA | TVA-Avg. | TQ-CC | TQ-BC | TQ-WE | TQ-Avg. | HERD  |  DQ  | Avg.  |
>   | - | :-: | :-: | :-: | :-: | :-: | :-: | :-: | :-: | :-: | :-: | :-: | :-: |
>   | Sora2 |  12s | 68.73 | 80.00 | 53.45 | 66.73  | 56.55 | 99.32 | 98.99 |  84.95  | 92.33 | 66.84 | 75.92 |
>   | Seedance 1.5 pro | 12s | 71.51 | 73.00  | 46.30 |  59.65  | 40.86 | 99.27 | 98.00 |  79.38  | 89.00 | 57.85 | 71.48 |
>   | Seedance 2.0     | 15s | 76.43 | 79.00  | 55.86  | 67.43  | 38.93 | 99.11 | 97.60 |  78.55  | 85.67 | 57.20 | 73.05 |
>   | Kling 3.0 |  15s | 76.50 | 71.00 |  60.47 | 65.74  | 22.32 | 99.26 | 99.79 |  73.78  | 87.33 | 57.62 | 72.19 |

---

> > ### Author Rebuttal · Reviewer_pxYm · 2026-04-03
> >
> > The author has addressed my concerns, so I am willing to increase my score to 5.

---

> > > ### Author Response · Authors · 2026-04-03
> > >
> > > Dear Reviewer pxYm,
> > >
> > > Thank you for the positive feedback and for increasing the score. We are pleased that our rebuttal addressed your concerns. We remain committed to incorporating the corresponding clarifications and discussions into the revised version of our paper.

---

### Official Review · Reviewer_Nn4N · 2026-02-28

**Soundness:** 2
**Presentation:** 3
**Significance:** 3
**Originality:** 2
**Overall Recommendation:** 3
**Confidence:** 3

**Summary:**

This paper introduces LoCoT2V-Bench, a benchmark for evaluating long-form text-to-video generation with complex prompts. The benchmark comprises 234 real-world videos with multi-scene prompts averaging 248.85 words, addressing a gap where existing benchmarks focus on short clips with simple prompts. The paper proposes LoCoT2V-Eval, a 5-dimensional evaluation framework covering perceptual quality, text-video alignment, temporal quality, dynamic quality, and HERD metric. Through evaluation of 13 models, the authors identify that current methods perform well on perceptual quality but struggle with fine-grained alignment and character consistency.

**Compliance With Llm Reviewing Policy:**

Affirmed.

**Key Questions For Authors:**

1. **On motivation and scope:** The benchmark defines "long-form and complex" primarily through prompt-side statistics (word count, complexity score). Could you clarify why intrinsic video properties (e.g., duration, shot transitions, camera dynamics) are not central to the problem definition? Relatedly, how do you justify the practical relevance of 248-word descriptive prompts when real-world text-to-video users typically provide much shorter, intent-driven inputs?

2. **On differentiation from VBench-Long:** VBench-Long already targets long text-to-video evaluation with a comprehensive framework. Beyond the prompt length comparison in Table 1, what are the concrete architectural and methodological differences that make LoCoT2V-Bench a necessary contribution? A detailed side-by-side comparison (shared metrics, unique capabilities, failure cases caught by one but not the other) would help us evaluate whether this benchmark offers sufficient novelty over existing work.

3. **On model selection and missing baselines:** All 13 evaluated models are open-source single-shot generators without narrative capability. Why were commercial systems with demonstrated long-form strengths (e.g., SeedDance 2.0, Kling 3.0, Sora 2, Veo 3) excluded? Furthermore, reporting the evaluation scores of the original real-world reference videos would provide a natural upper bound. Without these, it is difficult to assess where the field truly stands.

4. **On disentangling duration from complexity:** The current design simultaneously varies video length and prompt complexity. Could you provide ablation results (e.g., short+complex vs. long+simple prompts) that isolate whether the observed failures are caused by temporal coherence degradation over longer durations or by difficulty following detailed multi-scene instructions? This distinction is critical for guiding future model development and would strengthen the paper's contribution if addressed.

**Limitations:**

No, the authors didnt discuss the limitations in the submission.

**Strengths And Weaknesses:**

### Strengths

1. The paper is well-organized with a clear structure, making it easy to follow from problem motivation through benchmark construction and evaluation.

2. The paper provides extensive quantitative evaluations and case studies under the proposed 5-dimensional evaluation framework, offering a thorough empirical analysis across 13 models.

3. The idea of the HERD metric is interesting and has merit. Its dual-agent framework, which separates critique from scoring, outperforms both QA-based and direct scoring alternatives for capturing high-level narrative and emotional dimensions.


### Weaknesses

1. **Unclear motivation and problem definition.** The definition of "long-form and complex" text-to-video generation appears to focus primarily on the prompt side (word count, complexity score) rather than on intrinsic video properties (duration, shot transitions, camera movement, object dynamics, etc.), which limits the practical relevance of the benchmark. In particular, extending a prompt via an LLM is straightforward, yet the paper does not address the gap between such verbose descriptive language and the concise, intentional prompts that characterize real-world text-to-video workflows.

2. **Insufficient discussion of closely related work.** VBench-Long is also designed for long text-to-video generation and features a comprehensive evaluation framework. The authors should provide a substantive comparison discussing architectural and methodological differences rather than relying primarily on prompt word-length statistics in Table 1.

3. **Important details are missing.** What is the taxonomy of the 18 chosen themes, and how were they determined? Figure 6 indicates multiple rounds of human manual refinement, but the filtering statistics at each step are not reported. Concrete examples of filtered cases would help readers assess the curation quality.

4. **Ambiguous and narrow model selection.** The assertion that the 13 evaluated models are "representative" lacks justification, and the selection criteria are not explained. Moreover, all chosen methods are open-source single-shot generation models without narrative capability, which significantly limits the scope of the evaluation. We strongly suggest the authors include recent commercial systems (e.g., SeedDance 2.0, Kling 3.0, Sora 2, Veo 3) to provide a more complete landscape. Additionally, the evaluation scores of the original real-world videos should be reported as an upper-bound reference, allowing readers to gauge the gap between human-produced content, commercial products, and open-source models.

5. **Conflation of duration and complexity.** The paper simultaneously varies video duration (long-form) and prompt complexity, making it unclear whether the identified failures stem from temporal coherence challenges or from difficulty following detailed instructions. Ablations that separate these factors (e.g., short+complex vs. long+simple prompts) would clarify the true bottleneck.

---

> ### Author Rebuttal · Authors · 2026-03-31
>
> We thank you for your careful evaluation of our manuscript and your constructive comments. Our point-by-point responses are provided below. And we also provide some complementary results on [this site](https://anonymous.4open.science/r/LoCoT2V-Bench-1518/) (e.g., real-world video evaluation results and filtered video cases).
>
> **[W1/Q1: Motivation and Problem Definition]**
>
> **R1**: We appreciate the reviewer's feedback regarding the motivation and scope of our benchmark. We would like to clarify these as follows:
>
> * **Video-Centric Definition**: Our problem definition is not limited to the prompt side. "Long-form" is explicitly defined by video-side attributes: **>10s** duration and **multiple scenes**. Accordingly, **LoCoT2V-Eval** prioritizes intrinsic qualities (e.g., character consistency, temporal dynamics) over simple text matching.
> * **Professional Utility**: While casual users use short prompts, **professional production** (e.g., filmmaking) requires script-level control. We bridge the gap between "intent" and "execution" by testing models under complex, multi-attribute instructions.
>
> **[W2/Q2: Differentiation from VBench-Long]**
>
> **R2**: We value the reviewer's suggestion to further contrast our work with VBench-Long and will clarify the substantive architectural and methodological distinctions that make our LoCoT2V-Bench a necessary contribution below:
>
> * **Difference in Prompt Philosophy**: VBench-Long targets isolated attributes via short, single-scene prompts, while our LoCoT2V-Bench evaluates **professional narratives** involving complex multi-scene interactions and fine-grained visual details.
>
> * **Shared Metrics Comparison**: As shown below, we conducted side-by-side comparison across all shared dimensions and found that our LoCoT2V-Bench consistently achieves higher correlations (PLCC) with human preferences.
>
>   |Dimension|Perceptual Quality|Overall Alignment|Character Consistency|Background Consistency|
>   |-|:-:|:-:|:-:|:-:|
>   |VBench-Long|55.10|10.99|41.09|45.06|
>   |**LoCoT2V-Bench (ours.)**|**71.39**|**63.38**|**47.17**|**52.80**|
>
> **[W3: Lack of Important Details]**
>
> **R3**: We thank the reviewer for the suggestion. We clarify the systematic rigor of our data pipeline as follows:
>
> * **Taxonomy**: Our 18 themes were derived from trending tags on YouTube/Bilibili to ensure real-world coverage. We used keyword-based filtering via *yt-dlp* (Sec. 3.1) to ensure initial thematic relevance.
> * **Curation Statistics**: We started with **~1,000 videos**, applying a strict filtering process: (1) **~70% excluded** due to visual artifacts (watermarks, occlusions), low resolution, or misalignment; (2) Further exclusions due to MLLM encoding issues. This resulted in the 234 videos.
> * **Clarification on Fig. 6**: We clarify that Fig. 6 illustrates **prompt refinement**, not video filtering. Once the 234 videos were finalized, no samples were discarded; instead, they were just iteratively refined utill good enough.
>
> **[W4/Q3: Lack Evaluation on Commercial Methods and Real-world Videos]**
>
> **R4**: We appreciate the reviewer's constructive feedback on model selection and the suggestions to take the real-world reference videos as upper-bound. We explain these as follows:
>
> * **Commercial Baselines**: Please refer to our response **R2** to the **Reviewer pxYm** for a detailed discussion.
> * **Original Videos as Upper Bound**: They fail to serve as upper bound because (1) Real-world footage often contains motion blur, scoring lower in Perceptual Quality than "pure" AI outputs. (2) Our iterative expansion (Fig. 6) introduces complex narratives and specific characters absent from the original description so they would score poorly on metrics like Text-Video Alignment and Character Consistency. (3) Resolution of some source videos is too high to process (e.g., 4K).
>
> **[W5/Q4: Disentanglement of Video Duration and Prompt Complexity]**
>
> **R5**: We appreciate the reviewer's insightful suggestion to isolate these factors. We conducted a preliminary ablation on a representative subset (54 samples) due to time constraints, comparing **Long+Complex** (original, >30s) vs. **Short+Complex** (5–10s) using CausVid and LongLive.
>
> * **Experimental Analysis**: Interestingly, TVA and CC scores improve with longer duration. Shortening to 5–10s causes "content compression," where models fail to fully manifest the details defined in the prompts. While PQ improves in short clips, the drop in alignment (TVA) confirms that the primary challenge is **instruction following under structural complexity**, not just temporal decay.
>
>   |Model|PQ|TVA|TQ|HERD|DQ|
>   |-|:-:|:-:|:-:|:-:|:-:|
>   |CausVid|68.82|**37.04**|**80.50**|72.41|**58.43**|
>   | ~short-complex|**76.01**|32.02|79.43|**73.09**|51.04|
>   |LongLive|80.12|**48.62**|**84.59**|84.01|**62.11**|
>   | ~short-complex|**81.53**|41.19|81.56|**84.57**|53.89|

---

> > ### Author Rebuttal · Reviewer_Nn4N · 2026-04-03
> >
> > I appreciate the authors’ detailed rebuttal and the additional clarifications/ablations.
> > The response partially addresses my concerns, especially regarding dataset curation details and the preliminary duration-vs-complexity analysis. However, my main reservations remain: the benchmark’s practical motivation is still not fully convincing, and the differentiation from VBench-Long is only partially substantiated. We still have no idea what're the 18 themes.
> > Therefore, I keep my original recommendation of weak reject.

---

> > > ### Author Response · Authors · 2026-04-05
> > >
> > > Dear Reviewer Nn4N,
> > >
> > > We sincerely appreciate your constructive feedback. We would like to further address your concerns as follows:
> > >
> > > * **Motivation Restate**:
> > >   * **Problem Definition**
> > >     * Our definition of "long-form and complex" is not merely a statistical measure of word count, but a measure of semantic constraint density. Similar to how DPG-Bench [1] advanced T2I evaluation by shifting toward complex prompts, we believe the next frontier for T2V is handling diverse content under intricate, multi-layered textual instructions.
> > >     * As all know that LLMs can effortlessly generate "verbose" descriptions, the core challenge lies in the T2V model's ability to remain robust under such detailed guidance. Our benchmark explicitly tests whether models can faithfully realize video-side attributes—such as video dynamics and temporal consistency, and our prompts also contain related elements. Although certain properties like shot transitions remain an open challenge for evaluation, this study focuses on the multi-dimensional quality of videos generated from complex, real-world-style inputs.
> > >   * **Application Scenarios**
> > >     * We think that the gap between "concise" and "verbose" prompts is actually the gap between casual curiosity and professional production. In professional workflows (e.g., filmmaking), users often do not rely on the model's "random imagination." Instead, they use detailed scripts to "shrink" the generation space, ensuring the output aligns with a specific, intentional artistic vision (e.g., precise character settings and choreographed movements).
> > >     * Moreover, since text is currently the primary interface for T2V control, evaluating a model's ability to follow multi-constraint, coupled instructions is the most direct way to determine its utility for high-end, controllable content creation.
> > > * **Differentiation with VBench-Long on Prompt Design**: The core philosophy of VBench-Long is dimension-oriented while our LoCoT2V-Bench is content-oriented. Specifically, VBench-Long uses synthetic prompts to test isolated attributes (e.g., "a red car" for evaluating "color"). Our prompts are derived from real-world creator content, where attributes are inherently coupled (e.g., specific characters with multiple preset attributes performing complex actions or interact with each other among several scenes in detailed settings, **as our provided sample "soap_opera_4" in Appendix E**) instead of "atomic" capabilities by isolating specific attributes in VBench-Long. This is to evaluate model how to handle inter-attribute interference, a much more rigorous test for professional use.
> > > * **Methodological Differentiation**: Our LoCoT2V-Eval introduces substantive technical advancements over VBench-Long as fllows:
> > >   * **Finer-grained Alignment Evaluation**: The prompts are decomposed into multiple levels including scene, characters and so on for a diagnostic assessment, whereas VBench-Long primarily focuses on overall alignment.
> > >   * **Finer-grained Subject Consistency Evaluation**: Unlike VBench-Long's adjacent-frame DINO-v2 similarity (which can conflate background and subject), SAM3 + MLLM was employed to explicitly extract and track specific characters. This handles non-continuous appearances and aligns significantly better with human judgment as shown in the previous rebuttal.
> > >   * **Advanced Dynamics & Perception**: We incorporate **segment-level** and **video-level** dynamics (following DEVIL[2]) instead of only focusing on pixel-level and optical-flow-based motion. Furthermore, the HERD metric was introduced to assess dimensions like **emotional response** and **narrative flow**, which are critical to human perception but absent in existing benchmarks.
> > > * **Clarification on 18 Themes**: We apologize for the lack of the textual description of our taxonomy. As depicted **in the central pie chart of Fig.1**, our prompts are divided and nearly evenly distributed into 18 themes. We here provide a table for the theme-level distribution below for better demonstration:
> > >
> > >   |Theme Name|food|sports|local culture|minivlog|pets|street|technology|marine|space|
> > >   |-|:-:|:-:|:-:|:-:|:-:|:-:|:-:|:-: |:-:|
> > >   |Prompt Num.|16|15|14|14|14|13|13|15|14|
> > >
> > >   |wildlife|natural phenomena|landscape|human nature|sci-fi|film|animation|game|soap opera|
> > >   |:-:|:-:|:-:|:-:|:-:|:-:|:-:|:-:|:-:|
> > >   |14|13|10|6|15|13|12|12|11|
> > >
> > > Thank you again for your valuable and insightful comments. These comments will greatly improve the study. We will add these information into the revised version.
> > >
> > > ---
> > > **Reference**:
> > >
> > > [1] Hu et al. *ELLA: Equip Diffusion Models with LLM for Enhanced Semantic Alignment.* 2024.
> > >
> > > [2] Liao et al. *Evaluation of Text-to-Video Generation Models: A Dynamics Perspective.* 2024.

---

### Official Review · Reviewer_nuej · 2026-03-12

**Soundness:** 3
**Presentation:** 3
**Significance:** 4
**Originality:** 3
**Overall Recommendation:** 5
**Confidence:** 4

**Summary:**

This paper introduces LocoT2V-Bench, which is a benchmark specifically aimed at use cases where long-form video is generated from text. To do this, the authors collected 234 existing YouTube videos, spanning 18 themes, to construct complex prompts with metadata including character descriptions, backgrounds, and camera movement. This metadata benchmark was made using a large, cascading MLLM pipeline. In addition to the benchmark, the paper proposes an evaluation framework which is based on the following five evaluation dimensions: Perceptual Quality (PQ), Text-Video Alignment (TVA, with coarse and fine-grained sub-scores), Temporal Quality (TQ), Dynamic Quality (DQ), and a novel Human Expectation Realization Degree (HERD). Using the two in tandem, research found that a gap exists with models fine-grained ability in this area, but that models do better with strong perceptual quality and background consistency.

**Compliance With Llm Reviewing Policy:**

Affirmed.

**Final Justification:**

The paper is technically sound and gives clear, useful results for long form video evaluation.

- Originality: The benchmark, HERD metric, and VQA framework form a clear and meaningful contribution to an area that needs more work.
- Significance: The paper tackles an important gap in long form, multi scene video evaluation, and the results are useful even with some practical limits.
- Clarity: The paper is well organized and easy to follow, with clear figures and strong flow.
- Rebuttal: The authors addressed my main concerns well and gave clear answers on pipeline design, benchmark scale, and theoretical grounding.

**Key Questions For Authors:**

- With the limited 234 samples, would you be able to demonstrate that the model rankings are statistically generalizable?
- For the HERD dimensions, what frameworks in film theory or psychology helped guide the creation of these dimensions?
- Would you be able to speak to the generalizability of such a large pipeline? Would there be any cost/compute concerns?

**Limitations:**

The authors provide an Impact Statement covering data collection ethics and content filtering, which is appreciated. However, the paper lacks a dedicated technical limitations section addressing several important gaps. Specifically, the authors should discuss:

1. the risk of compounding errors in the cascading evaluation pipeline and what happens when early-stage models like SAM3 fail.
2. the computational and financial cost of running the full suite, given the number of heavy models involved.
3. the relatively small benchmark scale of 234 samples compared to prior benchmarks and what that means for the statistical reliability of rankings.

**Strengths And Weaknesses:**

- Soundness: The author proposes that the new pipeline should break prompts down into scene-level metadata for structured VQA evaluation, improving on the existing simple CLIP assessment scoring. The human alignment studies presented in this paper also validate the idea that automated metrics closely reflect human judgement
- Presentation: The paper is extremely well structured, with each section logically flowing to the next. The illustrations of Fig 1 and 2 help convey the complexity of the pipeline in a visually appealing way
-Significance: Evaluating narrative coherence and character consistency in long, multi-scene videos is an increasingly important problem that existing benchmarks largely ignore. The finding that character consistency falls below 40% for most models is a clear and useful result for the community.
- Originality: The VQA framework and Auditor Evaluator design for HERD are both novel concepts that push the envelope in long-form video evaluation. Further, the author’s idea to ground the information in real world videos gives the research more generalizability.


Weaknesses:
- Soundness: I have some concerns with the way that the pipeline is constructed: with the nature of it being a layered, cascading pipeline with many layers, any errors in an early layer would compound throughout the entire pipeline. Further, the benchmark only has 234 YouTube samples, which is smaller than VM Bench and VBench-Long which both have about 1000 samples. This underscores the generalizability in some sense compared to other standard benchmarks.
- Significance: Given that the pipeline runs SAM3, FG-CLIP2, RAFT optical flow, multiple MLLM calls, and GPT-5 API, there is likely a high compute cost. This limits the generalizability to teams who do not have the same resources as more resource liberal organizations. It would be interesting to discuss this in short to further understand the limitations and generalizability.
- Originality: The six HERD dimensions are defined by the authors without reference to any established framework in cognitive science, film theory, or human perception research. Grounding these dimensions in existing literature would significantly strengthen this contribution

---

> ### Author Rebuttal · Authors · 2026-03-31
>
> We sincerely appreciate your thoughtful feedback and the time you invested in reviewing our work. Below, we provide detailed responses to each of your comments.
>
> **[W1/Q1/L1/L3: Pipeline Soundness & Benchmark Scale]**
>
> **R1**: We fully understand the reviewer's concerns about the robustness of the evaluation pipeline and the scale of benchmark. We will address these concerns as follows:
>
> * **For Error Propagation Issues**: Our pipeline includes structural safeguards such as:
>   * **Gating & Anchoring:** FGA uses a scene-level "existence gate"; if a scene is not detected, the subtree is pruned to prevent hallucination propagation.
>   * **Dual-Check Verifier:** We use a strong MLLM to verify SAM3 trajectories against character metadata, filtering tracking artifacts. Besides, as shown in **Fig.2**, we specifically extracted **simplest concepts (e.g., man, dog)** instead of **complex description (e.g., a man wearing a blue hat and black jacket)** for sam3 to reduce the difficulty of its tracking, and we will filter out the sam3 results with low confidence based on the threshold set empirically.
>   * **Human-in-the-Loop:** Metadata (prompts/attributes) underwent multi-stage human verification to ensure consistency before evaluation.
> * **For Benchmark Scale Issues**: While our LoCoT2V-Bench has 234 samples, its **evaluation density** far exceeds prior work on these aspects:
>   * **Granularity:** With **6.0 scenes/sample** (in Fig.9), it provides **>1,400 scene-level points**.
>   * **Complexity:** Our prompts average **248.85 words** (Complexity: 8.70) vs. VBench-Long's 7.64 words (Complexity: 2.54). Moreover, construction and evaluation cost creates heavy constraints on further scaling. Specifically, high-fidelity long video generation is extremely resource-intensive (e.g., SkyReels-V2 takes 2-4 hours per video on 2$\times$H20) and multi-scene video generation always involve massive commercial LLM API usage such as Vlogger and VGoT, significantly increasing the burden of users in evaluation when facing a larger set of test samples.
> * **For Statistical Reliability Issues**: We confirmed generalizability through: (i) **High Human Alignment** (PLCC 71.39% for Perceptual Quality); (ii) **Cross-Category Consistency** in model rankings across "Daily Life", "Nature", and "Virtual" domains; (iii) **Diagnostic Significance**, where performance gaps (e.g., Background vs. Character consistency) are consistent across all 13 evaluated models.
>
> **[W2/Q3/L2: Compute Cost & Accessibility]**
>
> **R2**: We quite agree with the reviewer's consideration of the evaluation cost and we would like to clarify these points from the following aspects:
>
> * **One-time vs. Recurring Cost:** GPT-5/Seed-1.5 are used **only** for one-time offline benchmark construction. We will **release all structural metadata** (segmentation, attributes), so users can evaluate models without any API costs.
> * **Hardware Efficiency:** Our evaluation is **modular** like *VBench*[1] and *EvalCrafter*[2]. Models (SAM3, FG-CLIP2) are loaded sequentially, not simultaneously. Core metrics (BC, WE, DQ) use **streaming processing**, allowing execution on a single standard GPU (one H20 in our practice) without VRAM bottlenecks.
> * **Necessity:** Standard metrics (CLIP-score) cannot well capture long-term identity/narrative coherence; advanced tools are essential for meaningful LVG benchmarking. However, we are glad to explore more lightweight implementation to reduce compute cost in the future.
>
> **[W3/Q2: Theoretical Grounding of HERD]**
>
> **R3**: We sincerely thank the reviewer for this constructive feedback. We agree that grounding the HERD dimensions in existing literature further solidifies the theoretical foundation of our benchmark. Actually HERD dimensions are grounded in established **Cognitive Film Theory** and **Narratology**:
>
> * **Narrative Flow/Character Dev:** Based on *Classical Film Narratology* [3] and *Character Engagement Theory* [4] regarding "storyworld" coherence.
> * **Visual Style:** Maps to *Mise-en-scène*, analyzing how lighting/framing establish mood.
> * **Emotional Response:** Grounded in *Aesthetic Appraisal Models* [5] for psychological impact.
> * **Themes/Impression:** Rooted in *Semiotics* [6], assessing the synthesis of cinematic elements.
>
> We will add a formal subsection in our revised version to detail these mappings.
>
> ---
>
> **Reference**:
>
> [1] Huang et al. *VBench: Comprehensive Benchmark Suite for Video Generative Models.* 2024.
>
> [2] Liu et al. *EvalCrafter: Benchmarking and Evaluating Large Video Generation Models.* 2024.
>
> [3] Bordwell  D. *Narration in the Fiction Film.* 1985.
>
> [4] Smith M. *Engaging Characters: Fiction, Emotion, and the Cinema*. 1999.
>
> [5] Leder H et al. *A model of aesthetic appreciation and aesthetic judgments.* 2004.
>
> [6] Metz C. *Film Language: A Semiotics of the Cinema.* 1974.

---

> > ### Author Rebuttal · Reviewer_nuej · 2026-04-03
> >
> > Overall, the rebuttal increased my confidence and supports my recommendation. This is a solid contribution that others can build on. I therefore changed the overall recommendation to accept. The paper is technically sound and gives clear, useful results for long form video evaluation.

---

> > > ### Author Response · Authors · 2026-04-04
> > >
> > > Dear Reviewer nuej,
> > >
> > > Thank you for your further recognition of our work and for the score increase. We are glad that our response clarified your concerns. We remain committed to integrating all the points raised during this discussion into our revised version to enhance its clarity and depth.

---

### Decision · Program_Chairs · 2026-04-30

**Decision:**

Accept (regular)

**Comment:**

### Reasons to accept

The paper addresses a crucial data gap in building and evaluating long-form video understanding models by presenting both a dataset for long-form text-to-video generation with complex, multi-scene prompts, and a supporting evaluation framework with metrics for objective artifact detection and subjective quality assessment. The dataset is benchmarked with multiple T2V models, and an analysis of the capabilties and limitations of the models is provided as clear takeaways from the experiments. The reviews are also unanimous that the paper is generally well-structured and well-written.

### Reasons to reject

Reviewers raised concerns about the motivation behind the HERD metrics, the distinction from existing benchmarks such as VBench-Long, and the rationale for including or excluding specific T2V models in the benchmark. The authors have addressed these concerns in the rebuttal through additional explanations and experimental analyses. The main paper remains materially incomplete without these details, and the authors are strongly advised to add them to the main paper.

### Overall recommendation

The paper makes a solid contribution overall. The presented dataset and benchmarking details address fundamental aspects of video understanding, and are likely to be of interest to the wider audience at ICML.